# Semaphorin 3A causes immune suppression by inducing cytoskeletal paralysis in tumour-specific CD8+ T cells

Mike B. Barnkob [1,8] ✉, Yale S. Michaels [2,9,10], Violaine André[1], Philip S. Macklin [3], Uzi Gileadi [1], Salvatore Valvo [4], Margarida Rei [1,11], Corinna Kulicke [1,12], Ji-Li Chen[1], Vitul Jain [5], Victoria K. Woodcock [1], Huw Colin-York [1], Andreas V. Hadjinicolaou [1,13,14], Youxin Kong[5], Viveka Mayya[4], Julie M. Mazet [4], Gracie-Jennah Mead [4], Joshua A. Bull [6], Pramila Rijal[1], Christopher W. Pugh [3], Alain R. Townsend [1], Audrey Gérard [4], Lars R. Olsen [7], Marco Fritzsche[1,4], Tudor A. Fulga[2], Michael L. Dustin [4], E. Yvonne Jones [5] ✉ & Vincenzo Cerundolo[1,15]

Semaphorin-3A (SEMA3A) functions as a chemorepulsive signal during development and can affect T cells by altering their filamentous actin (F-actin) cytoskeleton. The exact extent of these effects on tumour-specific T cells are not completely understood. Here we demonstrate that Neuropilin-1 (NRP1) and Plexin-A1 and Plexin-A4 are upregulated on stimulated CD8+ T cells, allowing tumour-derived SEMA3A to inhibit T cell migration and assembly of the immunological synapse. Deletion of NRP1 in both CD4+ and CD8+ T cells enhance CD8+ T-cell infiltration into tumours and restricted tumour growth in animal models. Conversely, over-expression of SEMA3A inhibit CD8+ T-cell infiltration. We further show that SEMA3A affects CD8+ T cell F-actin, leading to inhibition of immune synapse formation and motility. Examining a clear cell renal cell carcinoma patient cohort, we find that *SEMA3A* expression is associated with reduced survival, and that T-cells appear trapped in SEMA3A rich regions. Our study establishes SEMA3A as an inhibitor of effector CD8+ T cell tumour infiltration, suggesting that blocking NRP1 could improve T cell function in tumours.

A major factor in anti-tumor immunity is the ability of T cells to infiltrate and function in a suppressive tumor microenvironment (TME)[1]. Whilst some tumors, known as immune deserts, lack T cell infiltration due to the absence of suitable antigens or defects in antigen presentation, other so-called immune restricted tumors utilize a combination of suppressive mechanisms in order to grow and metastasize[2]. These include recruitment of myeloid derived suppressor cells[3] and/or regulatory T cells as well as upregulation of inhibitory checkpoints such as programmed death-ligand 1 (PD-L1) and molecules including indoleamine 2,3-dioxygenase (IDO)[4] and transforming growth factor beta (TGFβ)[5]. It is also clear that structurally abnormal vasculature play an important role in restricting T cell infiltration by creating localized regions with low blood flow and hypoxia[6]. Absence of adhesion and immunomodulatory molecules such as intercellular adhesion molecule 1 (ICAM-1)[7] and upregulation of PD-L1[8] and Fas ligand[9] on endothelial cells can each inhibit T cell transmigration into the tumor parenchyma. As such, therapeutic interventions that allow T cell to ignore these signals represent an important unmet clinical need.

The cell-guidance systems normally associated with development have also been observed to affect leukocyte migration[10–12]. The secreted

protein semaphorin-3A (SEMA3A) is known to guide both endothelial cells and neurons during embryogenesis through the cell-surface receptor family Plexin-A[13,14]. SEMA3A binding to Plexin-A requires the co-receptor NRP1[15,16], which is found on CD4+ regulatory T cells (Treg cells)[17,18] and tumor-infiltrating CD8 T cells[19,20]. In axonal growth cones, SEMA3A signaling leads to profound changes in filamentous actin (F-actin) cytoskeletal organization[21], an effect that is thought to be dependent on myosin-IIA activity[22]. SEMA3A can also be produced by cancer cells[23] and recent evidence indicates that NRP1, like PD-1, is upregulated on dysfunctional tumor-specific CD8+ T cells and can modulate their anti-tumor response[19,24–26]. However, it remains contentious whether the SEMA3A-NRP1 axis is immunosuppressive[23,27] or supportive of CD8+ T cells' response to tumors[28]. Furthermore, due to the anti-angiogenic effects of SEMA3A[29], several groups have proposed utilizing SEMA3A to inhibit tumor growth[28,30]. We therefore decided to examine the role of SEMA3A in anti-tumor immunity more closely.

In this study, we report that NRP1 and Plexin-A1 and Plexin-A4 are upregulated following activation of CD8+ T cells corresponding to the level of TCR stimulation. Using T-cell-specific NRP1 knockout (KO) mice and genetic models of SEMA3A in cancer cells, we show that T cell expression of NRP1 and tumor cell expression of SEMA3A controls CD8+ T cell infiltration. We find SEMA3A expression in both cancer cells and blood and lymphatic endothelial cells within the TME. In vivo and in vitro experiments show that SEMA3A can strongly affect CD8+ T cell movement and migration, and formation of the immunological synapse, by paralyzing CD8+ T cells' F-actin cytoskeleton. In clear cell renal cell carcinoma (ccRCC) patients we find that NRP1+ T cells often express PD-1

and are trapped in SEMA3A-rich areas. Taken together, our study implicates a form of immune inhibition utilized by tumors, namely by directly affecting the cytoskeleton of tumor-infiltrating T cells.

## Results

### Tumor-specific CD8+ T cells upregulate NRP1 and Plexin-A1

To establish whether SEMA3A can affect CD8+ T cells, we first examined expression of its cognate receptor NRP1 on naive and stimulated T cells. NRP1 was upregulated on human NY-ESO-1-specific HLA-A2-restricted CD8+ T cells, and on mouse OT-I CD8+ T cells (OT-I T cells), upon stimulation with their cognate peptides, NY-ESO-1$_{157-165}$ and Ovalbumin$_{257-264}$ (Ova), respectively (Fig. 1a, b). Analysis of transcriptional data from the Immunological Genome Project Consortium[31] of naive and effector CD8+ T cells corroborated these findings (Supplementary Fig. 1A). We examined whole OT-I T-cell protein lysate and found that two NRP1 isoforms exist in mouse T cells, with the larger NRP1 protein being the dominant form following activation (Supplementary Fig. 1B). To examine NRP1 regulation in CD8+ T cells, we utilized antigenic Ova peptides with varying affinities for the OT-I TCR[32], namely SIINFEKL (N4), SIIQFEKL (Q4) and SIITFEKL (T4) and found that NRP1 expression was correlated with both peptide concentration and affinity of TCR engagement (Fig. 1c).

NRP1 is a co-receptor for a number of cell-surface receptors, including TGFβ receptors 1 and 2 (TGFβR1-2)[33], VEGF receptor 2 (VEGFR2)[34] and Plexin-A1, Plexin-A2, Plexin-A3 and Plexin-A4 receptors[35], and its function is highly dependent on the availability of these receptors for downstream signaling. We thus screened CD8+ OT-I

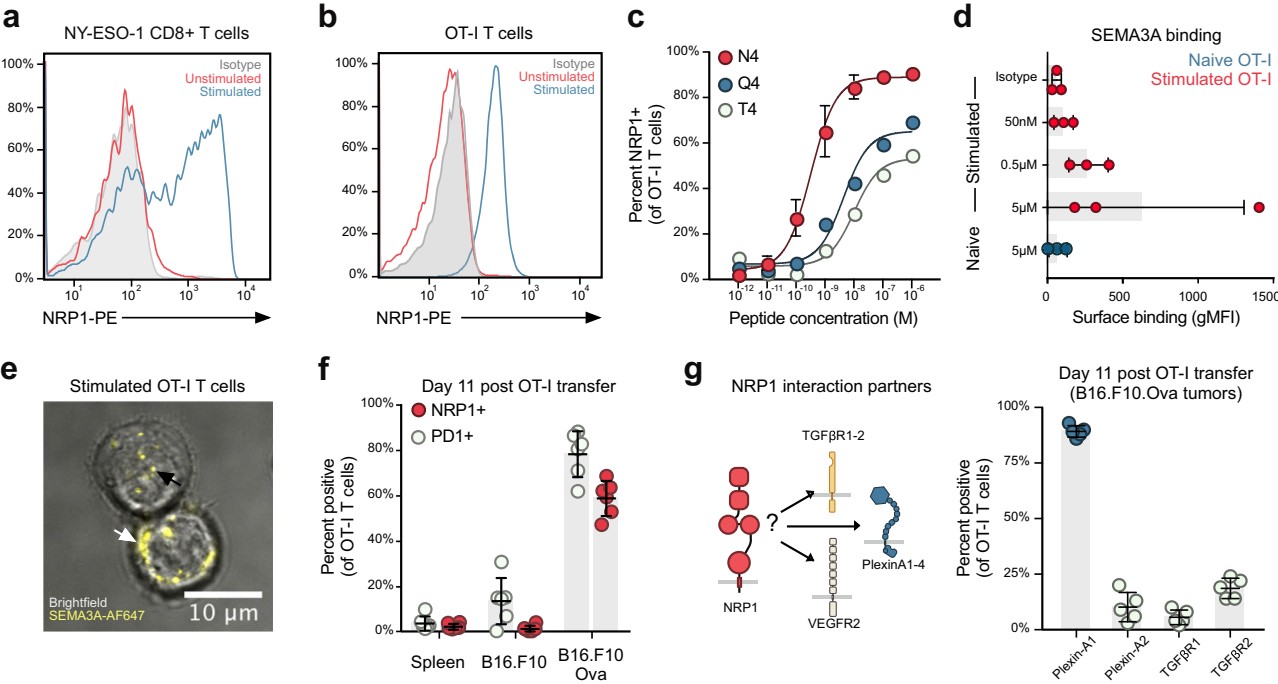

**Fig. 1 | Tumor-specific CD8+ T cells upregulate NRP1 and Plexin-A1 allowing for SEMA3A binding. a, b** Representative histogram of flow cytometric analysis of surface NRP1 expression on human NY-ESO-1-specific HLA-A2 restricted CD8+ T cells and mouse OT-I CD8+ T cells following 48 h stimulation with cognate peptides. Cells are gated on CD45, CD8 and TCRβ. Experiment repeated three times. **c** Analysis of NRP1 upregulation using peptides with varying TCR affinities. Cells are gated on CD45.1, CD8 and TCRβ. Cells from 3 mice per group, experiment was performed once. **d** Quantification of surface binding of SEMA3A$_{S-P}$ on naïve and 48 h stimulated OT-I T cells. Cells are gated on CD8 and CD3. Experiment was repeated three times. **e** Confocal imaging of 48 h stimulated OT-I T cells stained with AF647-labeled SEMA3A$_{S-P}$ shows that the protein can bind to the cell

membrane (white arrow) and within the cell (black arrow). Scale bar = 10 μm. Representative of two independent experiments. **f** Flow cytometric analysis of PD-1 and NRP1 expression on OT-I T cells 11 days after adoptive transfer in spleen, non-antigen expressing tumor (B16.F10) and antigen-expressing tumor (B16.F10.Ova) ($n = 6$ mice). Data representative of two independent experiments. **g** Schematic of NRP1 interactions partners (left). Flow cytometric analysis of expression of selected NRP1 interactions partners on OT-I T cells 11 days after adoptive transfer ($n = 5$ mice) (right). Experiment was performed once. Abbreviations: gMFI, geometric mean fluorescence intensity. N4, SIINFEKL. Q4, SIIQFEKL. T4, SIITFEKL. SD, standard deviation. Error bars are means ± SD (**c, d, f, g**) from representative experiments. Source data are provided as a Source Data file.

T cells for expression of NRP1 partner receptors. Stimulated, but not naive, OT-I T cells expressed Plexin-A1 but little to no Plexin-A2, TGFβR1, TGFβR2 or VEGFR2 (Supplementary Fig. 1D–F). Plexin-A4 was expressed at low levels on both unstimulated and stimulated cells. Having identified NRP1, Plexin-A1 and Plexin-A4 receptors on stimulated CD8+ T cells, we determined if these cells bind SEMA3A[16]. Indeed, flow cytometric analysis confirmed that only stimulated OT-I T cells could bind recombinant mouse SEMA3A$_{S-P}$, which includes the NRP1-binding Sema domain and the Plexin-Semaphorin-Integrin domain, but lacks the immunoglobulin-like domain and basic tail (Fig. 1d). Z-stacked, confocal imaging further indicated that SEMA3A$_{S-P}$ was internalized upon binding to T cells (Fig. 1e).

We next explored whether NRP1 and Plexin-A1 expression would persist on CD8+ T cells during infiltration in the TME. We adoptively transferred congenically marked and activated OT-I T cells into syngeneic C57BL/6 mice bearing B16.F10 and OVA expressing B16.F10 cells (B16.F10.Ova) in opposing flanks. While few OT-I T cells infiltrating B16.F10 control tumors were NRP1 positive, the majority of OT-I T cells residing within B16.F10.Ova tumors expressed NRP1 (Fig. 1f) and Plexin-A1 (Fig. 1g, right) up to 11 days after adoptive transfer. Of note, endogenous CD4+CD25+FoxP3+ T cells found within the tumor expressed both NRP1 and Plexin-A1 as well as TGFβR1-2 (Supplementary Fig. 1E), indicating that this subset of T cells might be modulated differently from CD8+ T cells. Collectively, these data show that NRP1 and Plexin-A1 receptors are upregulated on CD8+ T cells in a TCR-dependent manner, allowing recombinant SEMA3A to bind.

## NRP1-deficiency enhances anti-tumor activity of CD8+ T cells against SEMA3A rich tumors

To investigate the functional importance of SEMA3A in affecting T cell immunity in vivo, we used conditional knockout of *Nrp1*. Prior studies in the B16.F10 tumor model demonstrated that anti-NRP1 antibodies enhance CD8+ T cell infiltration and reduce tumor growth, but that knockout of NRP1 only in CD8 T cells had an effect only when combined with anti-PD-1 antibodies[26]. This suggested that NRP1 function on CD4 T cells might mask the effects NRP1 loss on CD8+ T cells. Therefore, we crossed LoxP-flanked (Flox) *Nrp1* mice with *Cd4^{Cre}* mice to generate *Cd4^{Cre}* X *Nrp1^{+/+}*, *Cd4^{Cre}* X *Nrp1^{Flox/+}* and *Cd4^{Cre}* X *Nrp1^{Flox/Flox}* mice (hereafter referred to as *Cd4^{Cre} Nrp1^{+/+}*, *Nrp1^{Flox/+}* and *Nrp1^{Flox/Flox}*, respectively), to generate NRP1-deficient T cells. Disruption of NRP1 expression on stimulated CD8+ T cells was confirmed by flow cytometric analysis (Supplementary Fig. 2A). Mice bred normally, had no gross anatomical differences, grew at similar rates and showed no sign of splenomegaly (Supplementary Fig. 2B, C). Analysis of thymocyte subsets and differentiated T cell memory subsets in the spleen revealed no differences between genotypes (Supplementary Fig. 2D, E), suggesting that NRP1 is not involved in thymocyte development or T cell homeostasis in non-inflamed conditions even when removed at the double-positive stage. CD8+ T cells from mice of all genotypes expressed similar levels of effector cytokines following CD3/CD28 stimulation (Supplementary Fig. 2F).

We then challenged *Cd4^{Cre} Nrp1^{+/+}*, *Nrp1^{Flox/+}* and *Nrp1^{Flox/Flox}* mice with B16.F10 and Lewis lung carcinoma (LL/2) cells. Notably, *Cd4^{Cre} Nrp1^{Flox/Flox}* mice had significantly delayed tumor growth and increased survival when challenged with either B16.F10 or LL/2 (Fig. 2a, b, Supplementary Fig. 2G). We confirmed that this effect was dependent on NRP1 deficient CD8+ T cells, as antibody-mediated depletion of CD8+ T cells allowed B16.F10 cells to grow unperturbed in *Cd4^{Cre} Nrp1^{Flox/Flox}* mice (Fig. 2c, Supplementary Fig. 2H). When examining levels of tumor-infiltrating lymphocytes (TILs) in *Cd4^{Cre} Nrp1^{+/+}*, *Nrp1^{Flox/+}* and *Nrp1^{Flox/Flox}* mice, we noticed a significant increase in the numbers of CD8+ T cells within tumors in *Cd4^{Cre} Nrp1^{Flox/Flox}* mice, but not of CD4+ T cells (Fig. 2d, Supplementary Fig. 2I). We wondered if the level of activation or exhaustion of CD8+ T cells was affected by NRP1 and could thus account for differences between genotypes, but found no

difference in PD-1, CD25, CD69 or CD44 expression levels on CD8+ T cells 15 days post tumor injection (Fig. 2e). Mixed bone marrow (BM) chimeric mice, containing *Cd4^{Cre} Nrp1^{Flox/+}* and *Nrp1^{Flox/Flox}* BM in lethally irradiated WT recipients, confirmed that the increased levels of infiltration were intrinsic to CD8+ T cells lacking NRP1 (Fig. 2f). We next set out to establish the role of NRP1 on CD8+ T cell priming and activation by infecting mice with the A/PR/8/34-derived pseudotyped influenza virus H7 (Netherlands/2003) N1 (England/2009) (here called H7N1 S-Flu). This virus is capable of triggering strong H-2 D^b-restricted influenza nucleoprotein (NP)-specific CD8+ T cell responses but, due to suppression of the hemagglutinin (HA) signal sequence, cannot replicate or generate anti-HA specific neutralizing antibodies[36]. This allowed us to specifically consider T cell responses. Mice were infected intranasally with H7N1 S-flu and weighed daily. No differences in weight between genotypes was observed (Supplementary Fig. 2J). We detected no differences in percentage or absolute number of H-2 D^b NP-tetramer positive CD8+ T cells in lungs, draining lymph nodes (dLN) or spleen, 10 days post-infection (Supplementary Fig. 2K, L). Examining the phenotype of CD8+ T cells in the lung, we found that infected mice from all genotypes had an expansion of effector T cells as compared to uninfected mice (Supplementary Fig. 2M). Thus, while NRP1 is dispensable for CD8+ T-cell priming and activation in flu-infected mice, the receptor plays an important role in anti-tumor responses.

We hypothesized that the reason T cell immunity was enhanced by NRP1-deficiency in our tumor models, but not against H7N1 S-flu, was an increased availability of SEMA3A in the former. Indeed, we did not find high levels of SEMA3A on either epithelial cells, leukocytes or endothelial cell-subsets in the lung before, during or after infection with H7N1 S-flu (Supplementary Fig. 3A, B). Conversely, aggressively growing tumors such as B16.F10, often generate a hypoxic TME[37] which itself can induce SEMA3A production[38]. We cultured B16.F10 cells in normoxic or hypoxic conditions and performed RT-qPCR. Hypoxic conditions led to upregulation of known hypoxic response genes, including *Pdk1*, *Bnip3* and *Vegfa*, in addition to upregulation of *Sema3a* transcript (Supplementary Fig. 3C). While not directly comparable to cells in the lung, flow cytometric analysis of B16.F10 cells grown for 11 days in vivo nonetheless showed expression of SEMA3A within the TME, mainly from tumor cells but also blood endothelial cells (BEC), lymphatic endothelial cells (LEC) and some CD45+ cells (Supplementary Fig. 3D, E). Immunofluorescence staining showed SEMA3A was mainly located around the core of the tumor, with SEMA3A rich areas containing many CD8+ T cells (Supplementary Fig. 3F). While multiple cell types thus contribute to SEMA3A production within the TME, we generated B16.F10.Ova cells with *Sema3a* knocked out or overexpressed (referred to as *Sema3a* KO and *Sema3a* OE, respectively) as the most straight forward way to experimentally manipulate the levels of SEMA3A levels in the TME. Deep-sequencing, RT-qPCR for *Sema3a* transcript and analysis by flow cytometry confirmed that cells lacked or over-expressed SEMA3A (Supplementary Fig. 3g, h). *Sema3a* OE and *Sema3a* KO cell lines grew at similar rates compared to wild-type B16.F10.Ova cells under both normal growth conditions and in the presence of the proinflammatory cytokines interferon gamma (IFNγ) and tumor necrosis factor alpha (TNF) in vitro (Supplementary Fig. 3I). Importantly, when we injected *Sema3a* OE and KO cell lines into opposite flanks of C57BL/6 mice, tumors grew at similar rates (Fig. 2g), thus confirming that the cell lines had a comparable growth potential in vivo. However, when we adoptively transferred stimulated OT-I T cells into these mice, *Sema3a* KO tumor growth was significantly delayed compared to *Sema3a* OE tumors (Fig. 2h). These results demonstrate that increasing SEMA3A in the TME through overexpression by tumor cells was sufficient to effectively suppress T cell mediated control of tumor growth. Furthermore, significantly fewer OT-I T cells had infiltrated B16.F10.Ova cells that overexpressed SEMA3A, compared to B16.F10.Ova cells that lack the ability to make SEMA3A (Fig. 2i). Taken together, our data underscores the functional significance of SEMA3A within the TME as a potent

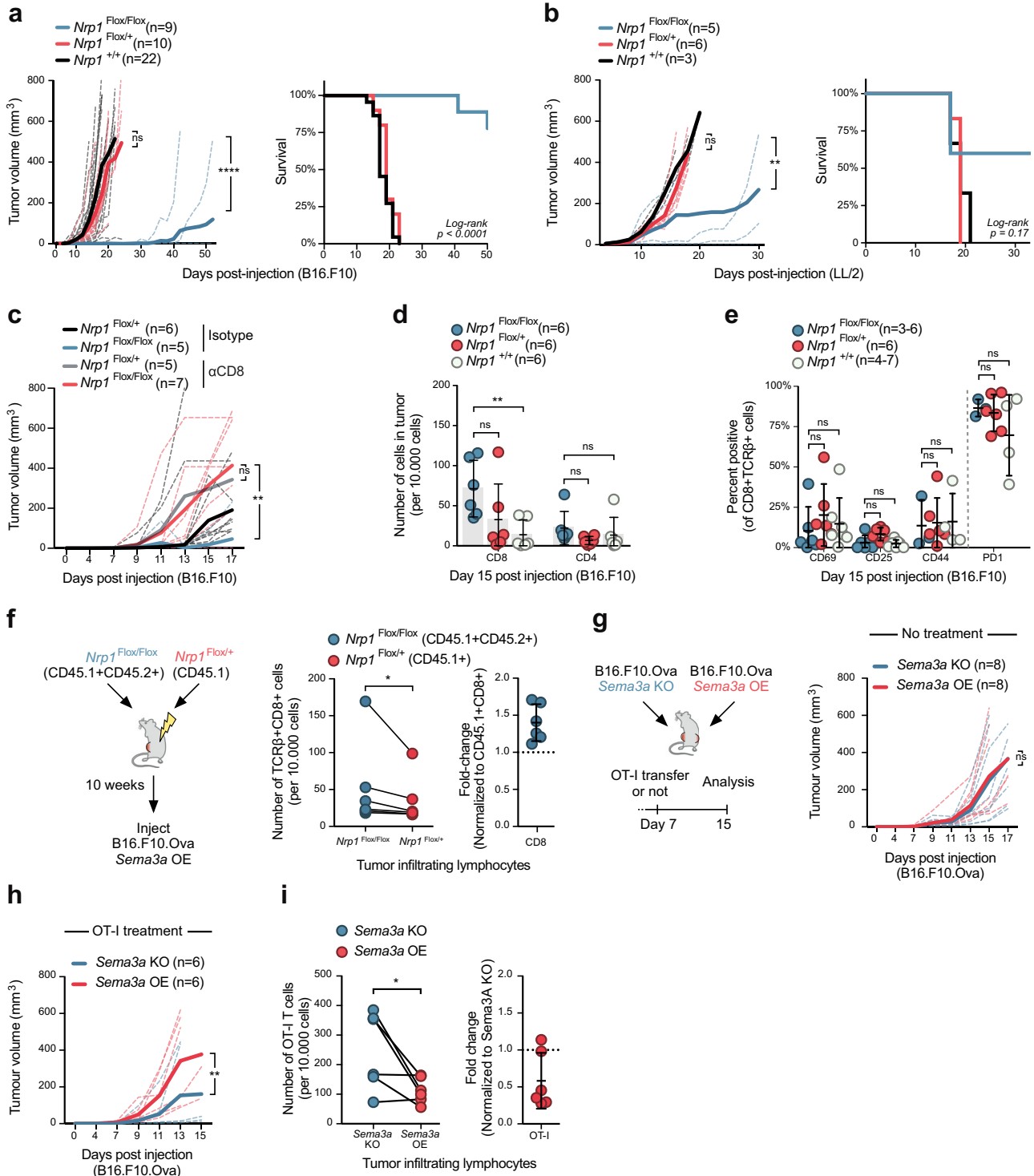

**Fig. 2 | NRP1-deficiency enhances anti-tumor migration and activity of CD8+ T cells. a** Growth curve of B16.F10 cells in *Cd4Cre Nrp1*+/+, *Nrp1*Flox/+ and *Nrp1*Flox/Flox mice (left) and Kaplan–Meier survival curve (right). Dashed lines indicate growth in individual mice, bold line average for group (*n* = 3–6 mice per group). **b** Growth curve of LL/2 cells in *Cd4Cre Nrp1*+/+, *Nrp1*Flox/+ and *Nrp1*Flox/Flox mice (left) and Kaplan–Meier survival curve (right). **c** Growth curve of B16.F10 cells in *Cd4Cre Nrp1*Flox/+ and *Nrp1*Flox/Flox mice pre-treated with either anti-CD8 antibody or isotype control. **d** Enumeration of CD4+ and CD8+ T cells infiltrated into B16.F10 tumors in *Cd4Cre Nrp1*+/+, *Nrp1*Flox/+ and *Nrp1*Flox/Flox mice (*n* = 6 per group). **e** Expression levels of activation and exhaustion markers on CD8+ T cells within the tumor 15 days post injection (*n* = 3–6 in *Nrp1*Flox/Flox group, 6 in *Nrp1*Flox/+ group and 4–7 in *Nrp1*+/+ group). **f** Experimental setup of mixed bone marrow chimeras in C57BL/6 mice (left) and subsequent enumeration of CD8+ T cells in mice (middle graph). Ratio of CD8+

T cells from *Cd4Cre Nrp1*Flox/Flox to *Nrp1*Flox/+ bone marrow derived cells (right graph) (*n* = 6 mice per group). **g** Experimental setup using B16.F10 *Sema3a* KO or *Sema3a* OE cells (left) and growth curve of cells in untreated mice (right) (*n* = 8 mice). **h** Growth curve of B16.F10 *Sema3a* KO or *Sema3a* OE cells using similar experimental setup as in (**f**), but with OT-I treatment at day 7 post-injection (*n* = 6 mice per group). **i** Enumeration of OT-I T cells in tumors (left graph) and their ratio of cells, normalized to the number in the B16.F10 *Sema3a* KO tumor (right) from same experiment as in (**g**). Abbreviations: ns, not significant. Error bars are means ± SD from 1 (**b**, **d–i**), or combined from 2 (**c**, **e**) and 4 (**a**) independent experiments. *\*P* = 0.0312 (**f**), *P* = 0.043 (**i**), *\*\*P* = 0.0041 (**b**), *P* = 0.0091 (**c**), *P* = 0.0072 (**d**), *P* = 0.008 (**h**), *\*\*\*\*P* < 0.0001 (**a**) by two-way ANOVA (**a–d**, **g**, **h**) or one-way ANOVA (**E**) or two-tailed paired *t*-test (**f**, **i**). Source data are provided as a Source Data file.

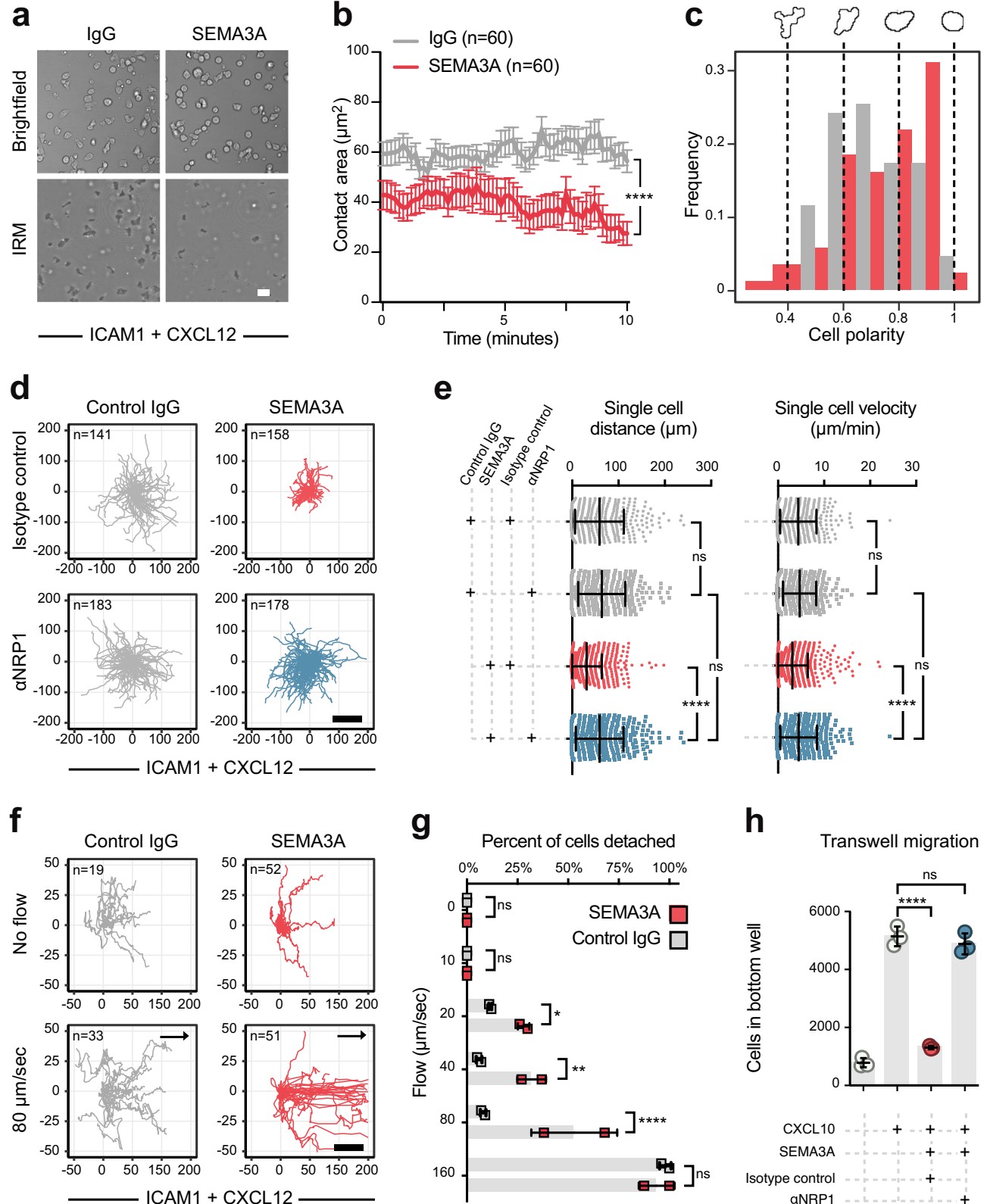

inhibitor of CD8[+] T cell migration, and thereby anti-tumor immunity, via interaction with NRP1.

## SEMA3A negatively regulates CD8[+] T cell adhesion, motility and chemotaxis through NRP1

Since in vivo experiments indicated that T cell migration was affected through NRP1, we undertook several experiments to dissect the effect of SEMA3A on CD8[+] T cell adhesion and motility. We first utilized interference reflection microscopy (IRM) to assess T cell adhesion[39]. This was done on plates coated with ICAM-1 and the chemokine ligand C-X-C motif chemokine ligand 12 (CXCL12/ stromal cell-derived factor 1 [SDF1]) in order to emulate the conditions found on endothelial cells and stromal cells within the TME[40]. When SEMA3A$_{S-P}$ was coated on plates, T cell adhesion was significantly weakened (Fig. 3a), an effect

**Fig. 3 | SEMA3A negatively regulates CD8⁺ T cell adhesion, motility and migration through NRP1. a** Representative brightfield and IRM images of stimulated OT-I T cells adhering to ICAM-1/CXCL12 coated plates with either SEMA3A$_{S-P}$ or IgG. Scale bar = 20 μm. **b** Contact area per single OT-I T cell. Representative of three independent experiments. **c** Frequency of cell polarity from brightfield images. A polarity of 1 indicates a shape of a perfect circle, 0 a rectangular shape. Representative images of OT-I T cells illustrated above graph. **d** Spider plots showing the migration paths of T cells pre-treated with either NRP1 blocking antibody or isotype control antibody on similar plates as in (**a**). Scale bar = 100 μm. **e** Single cell distance (left) and single cell velocity (right) from same experiment as in (**d**) (*n* = 380 cells for IgG and isotype control, 314 cells for IgG and anti-NRP1, 744 cells for Sema3A and isotype control, anf 403 for Sema3A and anti-NRP1). **f** Spider plots showing the migration path of OT-I T cells on ICAM-1/CXCL12 coated flow cells as in (**a**), with flow rates at 0 or 80 μm/s. Arrows indicate flow direction. Scale bar = 50 μm. **g** Percent of OT-I cells that detach in same experiment as (**f**) (*n* = 19 cells with control IgG and 73 cells with Sema3A). Representative of two independent experiments. **h** Number of stimulated OT-I T cells from individual mice (*n* = 3) able to transmigrate through 3 μm Boyden chamber in indicated conditions. OT-I T cells were pre-treated with either a blocking NRP1 antibody or isotype control antibody. Data representative of two independent experiments. Abbreviations: ns, not significant. Error bars are means ± SEM (**b**, **g**) or SD (**e**, **h**) from representative experiments (**b**, **h**) or combined from 2 (**g**) and 5 (**e**) independent experiments. *$P = 0.04$ (**g**), **$P = 0.003$ (**g**), ****$P < 0.0001$ (**b**, **e**, **g**, **h**) by two-tailed Student's *t*-test (**b**), two-way ANOVA (**e**, **g**) or one-way ANOVA (**h**). Source data are provided as a Source Data file.

that was present from initial attachment until at least 10 min later (Fig. 3b). In addition, T cells displayed a reduced polarized morphology (Fig. 3c, Supplementary Fig. 4a). T cell motility was also affected, as both distance and velocity were reduced when SEMA3A$_{S-P}$ was present, an effect that could be reverted by pre-treating T cells with a blocking anti-NRP1-antibody (Fig. 3d, e). Extravasation into tumors requires resistance of T cell adhesion to blood flow. To model this, we subjected T cells adhering to the ICAM-1 and CXCL12 coated surfaces as above to flow conditions like that in blood vessels. Whereas T cells treated with a control protein continued to adhere and migrate until flow velocities reached 160 μm/s, the SEMA3A$_{S-P}$ treated T cells mostly detached and were swept down-stream by 80 μm/s (Fig. 3f, g). Migration of T cells into the tumor parenchyma may also involve chemotaxis in response to gradients of chemokines such as CXCL10. Using a transwell assay, we found that SEMA3A$_{S-P}$ inhibited CXCL10-induced transmigration (Fig. 3h). We wondered if these effects were mediated through changed expression levels of integrins or selectin ligands involved in adhesion and extravasation. However, flow cytometric analysis did not reveal any down-regulation of CD11a (part of LFA-1), CD49d or CD162 (Supplementary Fig. 4b), suggesting that SEMA3A signaling does not affect expression of adhesion receptors on CD8⁺ T cells. These data illustrate that SEMA3A inhibit activated CD8⁺ T cell adhesion, motility and chemotaxis, effects that can be modulated using anti-NRP1-blocking antibodies.

## SEMA3A negatively regulates CD8⁺ T cell immunological synapse formation and cell-cell contact

Given the strong effects of SEMA3A on CD8⁺ T cell adhesion and motility, we also investigated whether SEMA3A also affects the formation of the immunological synapse (IS). We first tested the ability of CD8⁺ T cells to form close contacts with an activating surface displaying immobilized ICAM-1 and anti-CD3 antibodies. To mimic an environment in which SEMA3A had been secreted, T cells were added and allowed to settle in medium containing either SEMA3A$_{S-P}$ or control IgG, while the size and spreading speed of contact areas was measured using time-lapse IRM. T cells added to SEMA3A rich medium formed fewer and smaller contact zones (Fig. 4a. left, Supplementary Movie 1, 2). We noticed that cells in SEMA3A rich medium did not spread as much and were slower to adhere (Fig. 4a, right). Indeed, when analyzing contact zones over time, many cells in SEMA3A rich medium could not form large contact areas (Fig. 4b, top) and spread at a reduced velocity (Fig. 4b, bottom). These results were reminiscent of the effects seen when T cells were added to plates coated with ICAM-1, CXCL12 and SEMA3A$_{S-P}$ (Fig. 3a) and indicated that T cell ability to form IS could be compromised as well.

To examine the effects of SEMA3A on IS formation more closely, we utilized supported lipid bilayers containing ICAM-1, CD80 and H-2 K$^b$-Ova pMHC monomers. Stimulated OT-I T cells were pre-treated with fluorescently-labeled SEMA3A$_{S-P-I}$, which lacks the His tag that would lead to its interaction with the bilayer, to enable visualization of SEMA3A$_{S-P-I}$ binding cells, and visualized using time-lapse total internal reflection fluorescence (TIRF) microscopy. T cells with lower SEMA3A-binding formed classical IS containing a CD80-positive central supramolecular activation cluster (cSMAC) surrounded by an ICAM-1 rich peripheral supramolecular activation cluster (pSMAC), while T cells with higher SEMA3A$_{S-P-I}$ binding were unable to spread and appeared incapable of engaging with CD80 and ICAM-1 on the bilayer (Fig. 4c, Supplementary Movie 3). To quantify the extent of this defect, T cells were either left untreated, or treated with SEMA3A$_{P-S-I}$, allowed time to form IS if capable, fixed, washed to remove non-adherent cells and subjected to automated analysis of IS using a high throughput microscopy system[41]. Analysis of CD80 clustering and pSMACs formation indicated a 74% or 76% reduction, respectively, in IS formation in SEMA3A$_{S-P-I}$ treated T cells compared to untreated cells (Fig. 4d, e). The automated analysis was validated using non-cognate H-2 K$^b$-gp33 pMHC monomers on the bilayers leading to a false-discovery rate of <10% for CD80 clustering and less that 20% for pSMACs formation (Supplementary Fig. 5A). Among the SEMA3A-treated T cells that formed IS, there was a 20% reduction in CD80 accumulation and 17% decrease in the radial symmetry of IS (Supplementary Fig. 5B–D), indicating that even CD8⁺ T cells with low SEMA3A$_{P-S-I}$ binding, had impaired IS. We confirmed these findings by examining T cell binding to live cancer cells. Stimulated OT-I T cells and B16.F10.Ova cells were co-incubated in the presence of control IgG, SEMA3A$_{S-P}$ or a mutated SEMA3A protein, in which the NRP1 interaction site on SEMA3A has been mutated to substantially reduce the binding affinity[16], followed by enumeration of OT-I T cell:B16.F10.Ova cell-cell conjugates. There was a 50% reduction in conjugates in the presence of SEMA3A$_{S-P}$ compared to the untreated control or mutant SEMA3A treated conditions (Supplementary Fig. 5E). These results thus demonstrate that SEMA3A signaling leads to profound effects on the ability of CD8⁺ T cells to adhere to target cells and form an IS.

## SEMA3A severely affects T cell actin dynamics

Class 3 semaphorins have been shown to have various effects on the cytoskeleton in hematopoietic cells, including thymocytes[12], dendritic cells[42] and T cells[27], but the precise nature of these effects in CD8⁺ T cells is not well characterized. Since cytoskeletal F-actin remodeling is necessary for T cell binding to target cells[43], as well as lamellopodium[39], and IS formation[44,45], we examined F-actin content and dynamics in T cells during SEMA3A$_{S-P}$ exposure. We first treated stimulated OT-I T cells with SEMA3A$_{S-P}$ at varying durations and examined F-actin content using flow cytometry. Surprisingly, no significant differences in actin depolymerization were observed up to 30 min after SEMA3A$_{S-P}$ treatment (Fig. 5a). To better visualize F-actin dynamics before and after SEMA3A$_{S-P}$ treatment, we crossed LifeAct-eGFR[46] mice with OT-I mice to generate LifeAct-OT-I T cells. Stimulated T cells formed an active lamellopodium that undulated across an activating surface containing CD3 and ICAM-1 indicative of IS formation and allowing for close inspection of F-actin dynamics using time-lapse confocal microscopy. When SEMA3A$_{S-P}$ was added to cells during this activation phase, T cell morphology changed and took a more

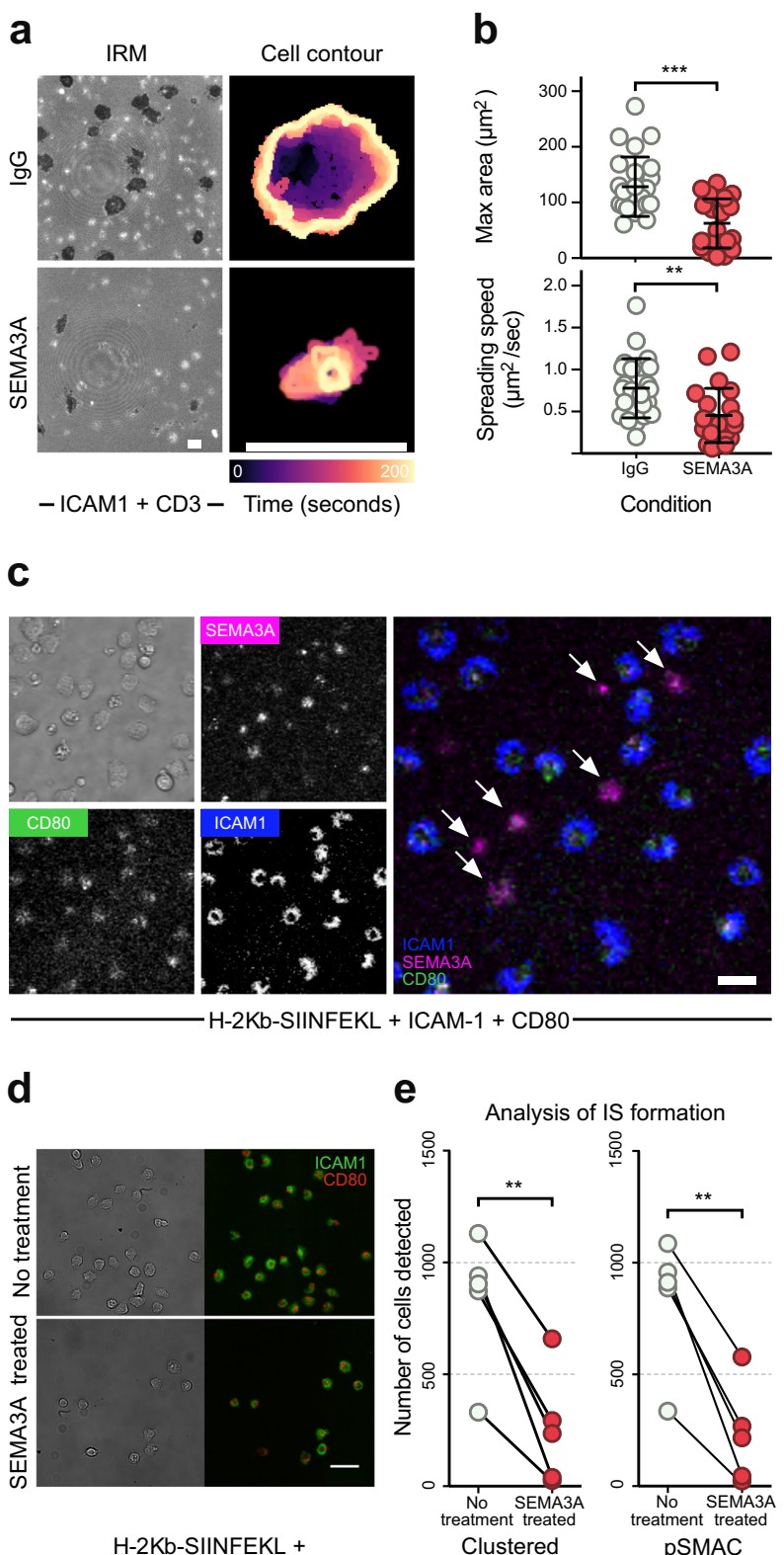

irregular and roughened appearance (Fig. 5b). During this phase, F-actin content at the surface interface did not change, but lamello-podia formation stopped and F-actin became non-dynamic and immobile (Fig. 5c, f, Supplementary Movie 4). To quantify the changes in F-actin turnover, we thus analyzed F-actin velocity along the lamel-lipodial cell edge using kymographs (Fig. 5d). SEMA3a$_{S-P}$ profoundly inhibited F-actin dynamics (mean velocity was 1.34 µm/min after treatment versus 3.8 µm/min before) (Fig. 5e), suggesting that F-actin turnover slowed down compared to control. In contrast, treatment of T cells with mutant SEMA3A, led to no significant differences in F-actin dynamics after treatment (Fig. 5e), confirming that the effect of SEMA3A on F-actin in the lamellopodia is NRP1-dependent. Next, we assessed whether this effect was due to localized F-actin depolymer-ization at the interface. Consistent with our flow cytometric analysis of

**Fig. 4 | SEMA3A negatively regulates CD8+ T cell immunological synapse formation. a** Live-cell imaging visualizing surface interface using IRM of stimulated CD8+ T cells dropped onto an activating surface with immobilized ICAM-1 and CD3 and SEMA3A$_{S-P}$ or IgG present in medium (left). Cell contour of representative cells from either condition (right). Color of contour indicates time from 0 to 200 sec as denoted on colorbar. Scale bar = 10 μm. **b** Quantification of maximum size of cell contact area (top) and spreading speed from initial contact to maximum contact area (bottom) (*n* = 25 cells per group) in same experiment as (**a**). **c** Live-cell imaging of activated T cells pre-treated with SEMA3A$_{S-P-I}$-AF647 and allowed to form synapses on supported lipid bilayers with ICAM-1, CD80 and H-2Kb-SIINFEKL. Arrows in merged image indicate cells that have bound SEMA3A and do not form

immunological synapses. Scale bar = 10 μm. Experiment performed once. **d** Representative image from high-throughput analysis of immunological synapses on supported lipid bilayers as in (**c**) with OT-I T cells pre-treated with SEMA3A or not. Scale bar = 30 μm. **e** Quantification of immunological synapses with or without SEMA3A$_{S-P-I}$ pre-treated OT-I T cells. (*n* = 90–1100 cells per mouse per group). Abbreviations: IRM, interference reflection microscopy. Sec seconds. Error bars are means ± SD combined from 3 (**b**) and 6 (**e**) independent experiments. **P = 0.0011 (**b**), *P* = 0.0042 (E, left), *P* = 0.0039 (E, right), ***P = 0.0002 (B) by two-tailed Mann–Whitney test (**b**) or two-tailed paired *t*-test (**e**). Source data are provided as a Source Data file.

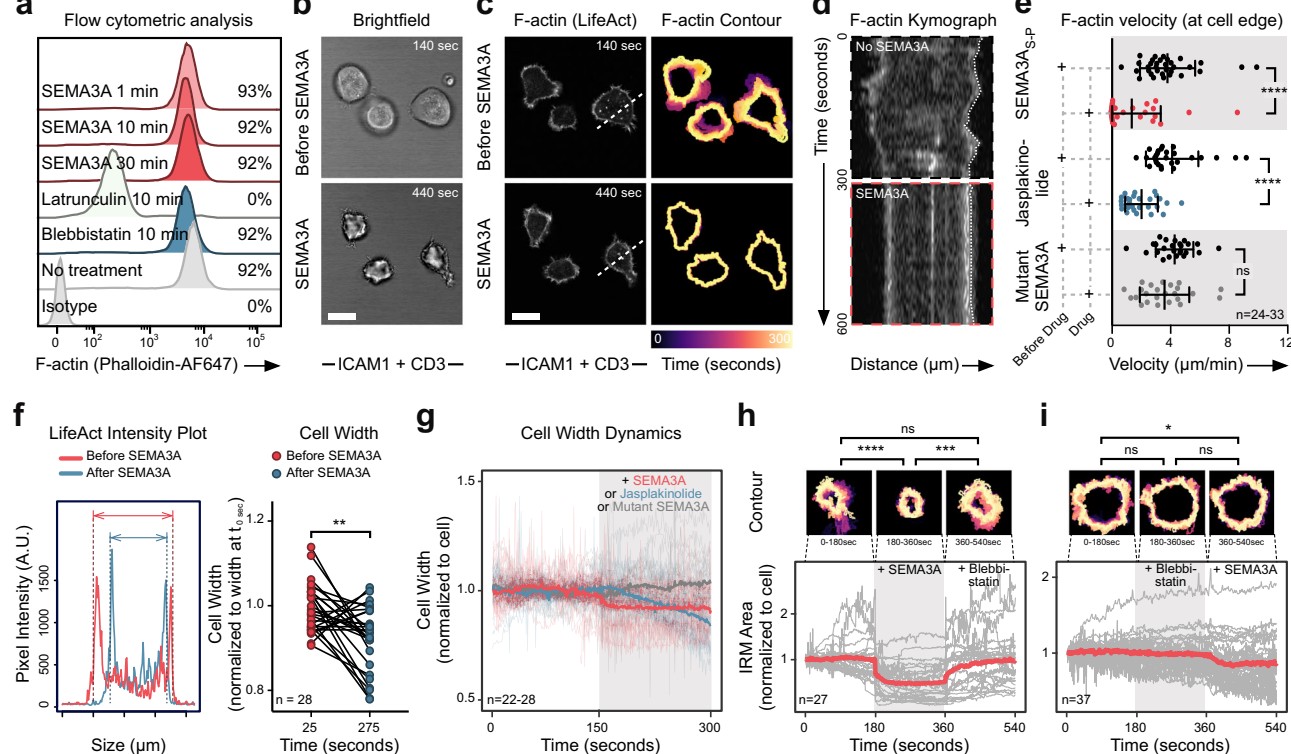

**Fig. 5 | SEMA3A affects T cell actin dynamics through actomyosin II.**
**a** Representative flow cytometric analysis of F-actin content with varying exposure to SEMA3A$_{S-P}$ in stimulated OT-I T cells. **b, c** Representative brightfield (**b**) or confocal (**c**) images of stimulated LifeAct OT-I T cells adhering to ICAM-1/CD3 coated plates before and after SEMA3A$_{S-P}$ added to medium. Color of contour (**c**–right) indicates time from 0 to 300 s. Dashed white line indicate area used for (**d**). Scale bar = 10 μm. Independently performed three times. **d** Kymograph before (top) and after (bottom) SEMA3A$_{S-P}$ added. Dotted line denote data used for calculating (**e**). **e** F-actin velocity at cell edge before and after treatment with either SEMA3A$_{S-P}$, Jasplakinolide or mutant SEMA3A (*n* = 33 cells in SEMA3A$_{S-P}$ group, *n* = 27 in Jasplakinolide group, and *n* = 24 in mutant SEMA3A group). **f** Intensity plot of LifeAct signal before and after SEMA3A$_{S-P}$ treatment of a OT-I T cell (left) or multiple cells exposed to SEMAA$_{S-P}$ (right) (*n* = 28 cells). **g** Cell width dynamics like

(**f**) before (white background) or after (gray background) SEMA3A$_{S-P}$, Jasplakinolide or mutant SEMA3A addition. **h** IRM area of OT-I T cells (gray lines) and average (red line) over time, with no treatment (leftmost white background), under treatment with SEMA3A (gray background) and Blebbistatin (rightmost white background). Representative contour plots of single cell under different treatments above, with color denoting time (150 sec total). **i** IRM area of individual OT-I T cells and contour plots as in (**h**), but with treatment with Blebbistatin (gray background) before SEMA3A$_{S-P}$ (rightmost white background). Abbreviations: A.U. arbitrary units, Min minutes, ns, not significant. Sec seconds, *t* time. Error bars are means ± SD from 1 (**f**) or combined from 2 (**a**) and 3 (**e, h, i**) independent experiments. *P = 0.016 (**i**), **P = 0.0011 (**f**), ***P < 0.0002 (H), ****P < 0.0001 (**e, h**) by two-tailed paired *t*-test (**f**) or Students t-test at time-points 90, 270 and 450 sec (**h, i**). Source data are provided as a Source Data file.

global F-actin abundance (Fig. 5a), the fluorescence intensity of LifeAct at the interface did not change, although the F-actin network contracted, and the cell width shrank substantially following treatment with SEMA3A$_{S-P}$ (Fig. 5f, g). Because these effects on the actin cytoskeleton suggested that F-actin turnover dynamics could be affected, we tested whether Jasplakinolide treatment would phenocopy the effects of SEMA3A$_{S-PI}$. In contrast to our expectations, this instead led to a shrinkage of the cells' F-actin network, not the immobilizing effects treatment with SEMA3A produced (Fig. 5g).

SEMA3A signaling through Plexin-A1 and Plexin-A4 inactivates the small GTPase Rap1A[47], which in turn can modulate myosin-IIA activity in diverse cell types[48,49]. The effects on the T cell cytoskeleton we observed in the presence of SEMA3A$_{S-P}$ appeared consistent with increased myosin-IIA activity. We therefore visualized and quantified the contact area of undulating T cells before and after SEMA3A$_{S-P}$ treatment followed by treatment with the myosin-II inhibitor Blebbistatin. As the border of IRM and F-actin signal overlay completely (Supplementary Fig. 5F), we quantified IRM area to avoid phototoxic

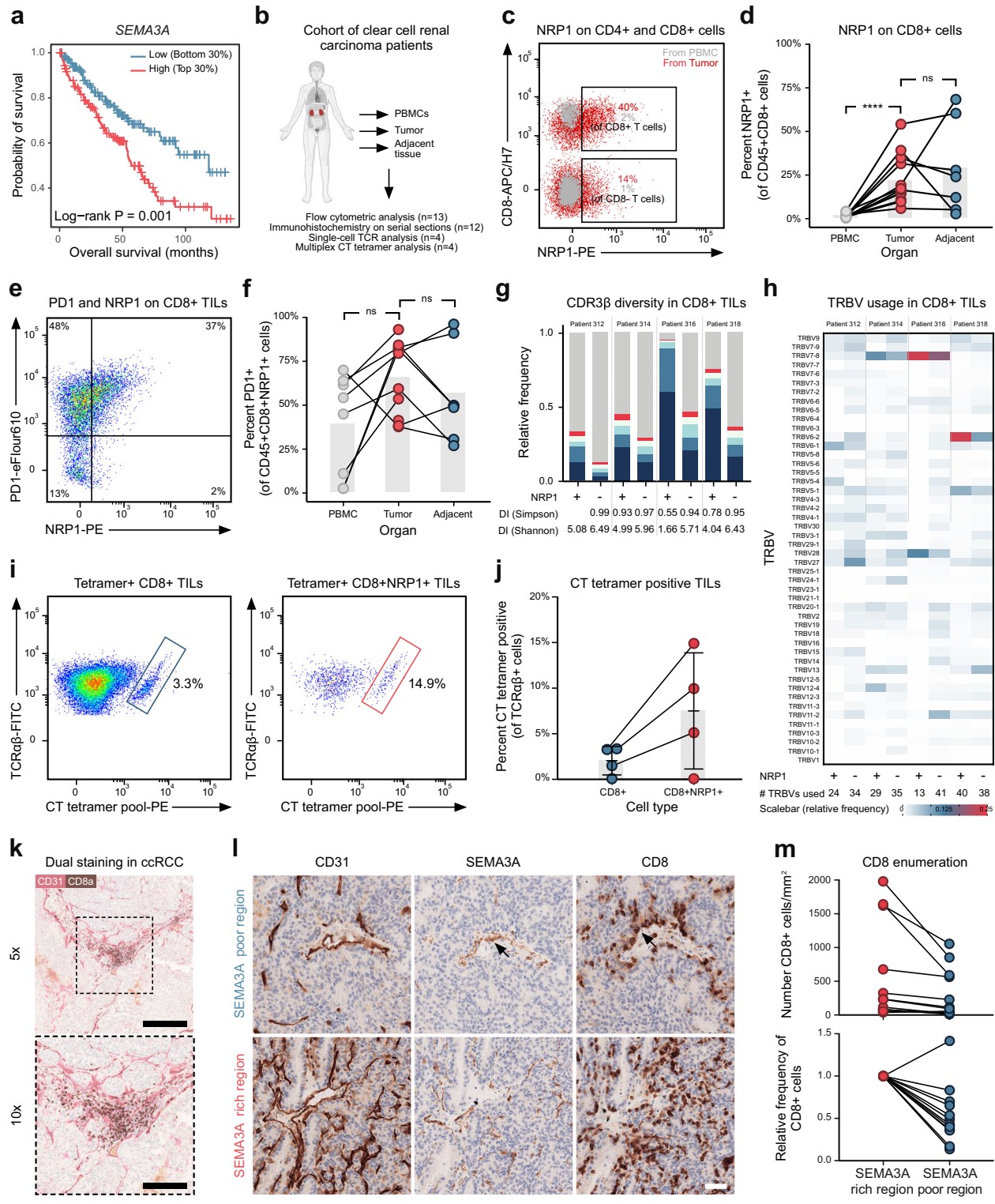

effects and inactivation of Blebbistatin, which would be caused by exciting LifeAct[50]. When SEMA3A$_{S-P}$ was added, T cell contact area contracted significantly and cells became immobilized, in line with our analysis of F-actin (Fig. 5f, g). In contrast, when Blebbistatin was added, T cells started undulating and regained their former contact size (Fig. 5h, Supplementary Movie 5). Conversely, when cells were pre-treated with Blebbistatin followed by SEMA3A$_{S-P}$, they retained their shape and activity (Fig. 5i, Supplementary Movie 6). We therefore speculate that SEMA3A inhibit F-actin dynamics in CD8⁺ T cells through hyper-activation of myosin-IIA.

**NRP1 is expressed on tumor-infiltrating CD8⁺ T cells in clear cell renal cell carcinoma patients**

We next wished to explore if our findings were relevant to human cancer. Analysis of publicly available TCGA data revealed that high *SEMA3A* expression was associated with poorer survival in clear cell renal cell carcinoma (ccRCC) (Fig. 6a), while another Semaphorin, *SEMA4A* was not (Supplementary Fig. 6A). SEMA3A expression was also associated with poor survival in cervix and low grad gliomas (Supplementary Fig. 6B). We hence turned to a cohort of ccRCC patients that had undergone nephrectomy (Supplementary Table 1) to explore the

**Fig. 6 | CD8⁺ TILs express NRP1 and are captured in SEMA3 rich areas in ccRCC tumors. a** Correlation of *SEMA3A* mRNA level with survival of ccRCC patients. **b** Schematic representing ccRCC patient cohort utilized in (**c**−**m**). **c** Representative flow cytometric analysis of CD8 and NRP1 expression in PBMC and TILs. **d** NRP1 expression on CD8⁺ T cells in PBMC, tumor and tumor-adjacent tissue (*n* = 12 samples from PBMC, 13 from tumor and 7 from tumor-adjacent tissue). **e** Representative flow cytometric analysis of PD1 and NRP1 on CD8⁺ TIL. **f** PD1 expression on NRP1 positive CD8⁺ T cells in PBMC, tumor and tumor-adjacent tissue (*n* = 8 samples from PBMC, 10 from tumor and 7 from tumor-adjacent tissue). **g** CDR3β diversity in NRP1 positive (⁺) and negative (-) CD8⁺ TILs (*n* = 4). Colored bars represent the five most abundant clonotypes, gray bars the remaining sequences. **h** Heatmap of TRBV usage in NRP1 positive (⁺) and negative (-) CD8⁺ TILs (*n* = 4). Color indicates relative usage within all of individual patients.

**i** Representative flow cytometric analysis of TCRαβ and CT tetramer positive CD8⁺ TILs (left) and NRP1⁺ CD8⁺ TILs (right). **j** Percentage CT tetramer positive NRP1⁺ and NRP1- CD8⁺ TILs. **k** Representative CD8 and CD31 staining in ccRCC. Dashed area indicates zoom area in bottom image. Scale bar = 500 μm (top) and 250 μm (bottom). **l** Representative CD31, SEMA3A and CD8 staining in SEMA3A poor region (top row) and SEMA3A rich region (bottom row). Arrows indicate association between SEMA3A and CD8 staining. Scale bar = 50 μm. **m** Enumeration of CD8⁺ TILs in SEMA3A rich and poor regions (*n* = 12). Abbreviations: CT cancer testis, DI diversity indices. ns not significant. SA Shannon index, SI Simpson index. TIL tumor-infiltrating lymphocytes. TCGA The Cancer Genome Atlas, TRBV TCR beta chain variable. Error bars are means ± SD. ****$P < 0.0001$ by two-way ANOVA (**d**, **f**). Source data are provided as a Source Data file.

role of the SEMA3A/NRP1 pathway in human cancer immunity (Fig. 6b). We first quantified NRP1 expression on CD8⁺ T cells from peripheral blood (PBMCs) and CD8⁺ TILs within tumor and tumor-adjacent tissue. Significantly more CD8⁺ T cells in both tumor and tumor-adjacent tissue expressed NRP1 (Fig. 6c, d), suggesting that these cells would be sensitive to SEMA3A. In our mouse model, NRP1 expression correlated with antigen exposure (Fig. 1c), and we therefore speculated that NRP1-positive CD8⁺ TILs might be tumor-specific. Indeed, most NRP1⁺ TILs were also PD-1-positive (Fig. 6e, f), demonstrating that they had either recently been activated or experienced chronic exposure to antigen[51]. We single-cell sorted NRP1-negative and positive CD8⁺ T cells from four ccRCC patients and examined their TCR repertoire. TCR diversity, as calculated by either Shannon (SA) and Simpson (SI) diversity indices of CDR3β (Fig. 6g) and TRBV usage (Fig. 6h), showed that NRP1⁺ CD8⁺ TILs were more clonal than NRP1- T cells, further supporting the hypothesis that such T cells had undergone clonal expansion following recognition of their cognate antigens[52] (Fig. 6e, f). Various cancer-testis (CT) antigens can be expressed by neoplastic cells in ccRCC[53]. We took advantage of this fact and screened four HLA-A2-positive patients for the presence of HLA-A2-restricted CT-antigen-specific CD8⁺ T cells using a panel of 21 HLA-A2 tetramers loaded with CT epitopes[54]. In the three patients who had CT tetramer positive TILs, we found that a larger proportion of NRP1⁺ CD8⁺ TILs were specific for CT-antigens (Fig. 6i, j, Supplementary Fig. 6C). Taken together these data show that NRP1⁺ CD8⁺ TILs were found in ccRCC patients, were activated and were likely specific for tumor-associated antigens.

We next explored the spatial distribution of SEMA3A and CD8⁺ T cells within the TME. For this purpose, we stained ccRCC tissue sections from 12 patients from our ccRCC cohort for SEMA3A by immunohistochemistry (IHC). We observed widespread expression of SEMA3A both within the tumor as well as in adjacent non-neoplastic kidney tissue. In the tumor, SEMA3A was predominantly expressed by smooth muscle cells within the tunica media of tumor vasculature but also in areas of fibromuscular stroma. In the adjacent tissue, glomerular mesangial cells and smooth muscle cells within peritubular capillaries stained positive for SEMA3A (Supplementary Fig. 6D). Next, strict serial sections from the same formalin-fixed paraffin embedded tissue blocks were stained for CD31 and CD8 and computationally aligned to the SEMA3A sections. Pathological review confirmed that expression of SEMA3A co-localized with that of the blood vessel marker CD31. Furthermore, CD8⁺ cells were often located within regions of high SEMA3A expression (Supplementary Fig. 6E); indeed dual-staining of CD31 and CD8 in ccRCC clearly showed that CD8⁺ cells are restricted to the immediate area surrounding blood vessels (Fig. 6k). To further explore the effect of SEMA3A on CD8⁺ cell infiltration and localization, we compared regions within each tumor that were either SEMA3A rich or SEMA3A poor, allowing us to control for variability in CD8⁺ cell infiltration between patients (Fig. 6l). This analysis confirmed that our selected SEMA3A rich regions expressed more CD31 than the SEMA3A poor regions, underscoring the close association of SEMA3A with the vasculature (Supplementary Fig. 6F, G). In 11

out of 12 examined patients, there were significantly more CD8⁺ TILs in the SEMA3A rich areas than in SEMA3A poor areas, corresponding to 46% fewer CD8⁺ cells in SEMA3A poor regions (Fig. 6m). Additionally, the CD8⁺ cells that were present in SEMA3A poor regions were often found clustered near sources of SEMA3A (Fig. 6l, arrows). We therefore suggest that SEMA3A trap CD8⁺ T cells following their infiltration to perivascular areas within the tumor.

## Discussion

In this study, we characterize the role of the secreted protein SEMA3A in controlling tumor-specific CD8⁺ T cells, highlighting important conclusions concerning its function.

SEMA3A is known to restrict neuronal migration[15], but can have opposing effects on immune cell motility. While both thymocyte[55] and macrophage[38] migration can be inhibited, SEMA3A has also been shown to increase dendritic cell (DC) migration[42]. Here we established several lines of evidence that reveal a strong inhibitory effect of SEMA3A on CD8⁺ T cell migration in tumors. First, in vitro experiments provided functional insights into how SEMA3A inhibited key steps in T cell extravasation, including adhesion, transmigration and mobility. Notably, these effects could be reversed using a blocking antibody against NRP1, confirming that NRP1 is an important regulator of SEMA3A signaling in CD8⁺ T cells. Second, conditional knockout of NRP1 on T cells corroborated these findings in different mouse cancer models, resulting in higher CD8⁺ T cell infiltration into the TME. Conversely, significantly fewer tumor-specific T cells homed to and infiltrated SEMA3A-overexpressing tumors. Third, in ccRCC patients CD8⁺ TILs were preferentially found in SEMA3A rich regions and beside Sema3 rich blood vessels, reminiscent of how tumor-associated macrophages can be entrapped within SEMA3A rich hypoxic regions[38]. While we do not functionally test these human TILs ability to migrate, their accumulation in SEMA3A rich areas are consistent with our in vitro finding of T cell paralysis and similar accumulation in mouse models. We thus hypothesize that NRP1⁺ T cells become stranded in these SEMA3A rich regions. These experiments are in line with findings from Leclerc et al., which also find that SEMA3A can inhibit T cell chemotaxis toward CXCL12 in transwell assays[19].

We also explored the effect of SEMA3A on IS formation. Previous studies have characterized SEMA3A as an inhibitor of T cell signaling and proliferation using in vitro assays[23,27]. We extended these results and confirmed that key steps in synapse formation are affected, including cell-cell binding, formation of close contact zones and organization of distinct supramolecular activation clusters. These findings are in line with work by Ueda et al. who found that SEMA3E inhibited IS formation in thymocytes[12]. We also show that the F-actin cytoskeleton becomes activated following SEMA3A exposure. Although further experiments are warranted to draw firm conclusions, this effect is ostensibly dependent on myosin-IIA activity, since we could rescue T-cell undulation using the drug Blebbistatin, which specifically prevents myosin-II activity[56]. High resolution 3D imaging has shown that myosin-IIA forms bona fide arcs above the pSMAC[57,58]

but then moves inwards and contracts, thereby pinching the T cell away during termination of the IS[39]. This isotropic contraction of the actomyosin arc appears similar to myosin's role during cytokinesis[59]. Our data suggest that SEMA3A leads to hyperactivation of myosin-II, thus enforcing IS termination. Data do exist to provide a link between SEMA3A binding and myosin-II. Biochemical and crystallographic studies have shown that SEMA3A signaling converts the small GTPase Rap1A from its active GTP-bound state, to its inactive GDP-bound state following binding to Plexin-A's[30,47]. In epithelial and endothelial cells, active Rap1-GTP can act as a negative regulator of myosin-II[48]. It is therefore tempting to speculate that SEMA3A, by inhibiting Rap1-GTP activity, leads to hyperactivation of myosin-II. Indeed, in both neurons[22,60] and DCs[42], SEMA3A has been shown to increase myosin-II activity in line with this interpretation; however, much of this pathway needs to be further elucidated in T cells. We propose that SEMA3A induces a cellular "paralysis" based on integrin-actinomyosin contraction leading to motility paralysis and immunological synapse preemption.

The source of SEMA3A in the TME are likely multiple. We show here that SEMA3A can be expressed by BEC and LEC cells in tumors, by some immune cells (Supplementary Fig. 3D, E) and also by the tumor cells themselves conditions (Supplementary Fig. 3C and E, F). Conversely, little to no expression was seen in draining and non-draining lymph nodes. Some immune cells, especially CD4+ T cells and dendritic cells, have previously been shown to upregulate SEMA3A when activated[27]. SEMA3A is also known to be expressed by endothelial cells during angiogenesis[13] where it plays an important autocrine role in vascular formation by regulating endothelial integrins[61]. SEMA3A might thus be upregulated as neo-angiogenesis happens in the tumor or as an attempt to normalize the vasculature[29].

There is a growing interest in NRP1 in the context of T cell anti-tumor immunity. Much research has focused on Treg cells, as NRP1 can be used to identify thymus-derived regulatory T cells[17] and has been shown to play an important role in controlling Treg cell function and survival[62]. It has become clear that NRP1 is expressed on dysfunctional tumor-specific CD8+ T cells[19,24,25], indicating that the protein might play an important role in regulating CD8+ T cells as well. We show that initial NRP1 expression is controlled by the level of TCR-engagement, that the protein remains expressed on tumor-specific CD8+ T cells in vivo and that NRP1 is found on a subset of tumor-specific CD8+ T cells in human ccRCC patients. These latter results are in line with reports by Jackson et al.[63] and Leclerc et al.[19] who found that ~10% of CD8+ TILs from melanoma patients and 14% of non-small-cell lung cancer patients, respectively, were NRP1 positive. Unlike us, Jackson et al. did not find any role for NRP1 in regulating CD8+ T cells when mice were challenged with a leukemia cell line. An explanation for this discrepancy could be a difference in SEMA3A expression between models. Indeed, we did not find any functional differences between NRP1 knockout and wild-type T cells when challenging mice with H7N1 S-Flu, a pathogen that did not lead to any meaningful upregulation of SEMA3A in the lung. Conversely, SEMA3A knockout or overexpression in B16.F10.Ova cells was shown to have significant effects on T cell migration and control of tumor growth when treated with tumor-specific CD8+ T cells. An even stronger effect was seen in the lack of tumor growth in our $Cd4^{Cre}$ $Nrp1^{Flox/Flox}$ mice. These results are in line with Delgoffe et al.[62] and Leclerc et al.[19] who show similar control of tumor growth when treating mice with a blocking anti-NRP1 antibody. Hansen et al. also found strong anti-tumor effects in a comparable conditional CD4[Cre] Nrp1 knockout model and ascribed this primarily to decreased Treg infiltration into the TME[64]. However, using a $E8I^{Cre}$-NRP1 model, to specifically remove NRP1 on mature CD8+ T cells, Liu et al., only found an effect on tumor growth, if an anti-PD1 antibody was used co-committedly[26]. Why do we and Hansen et al.[64] see a strong effect on primary tumor growth, while Liu et al.[26] did not? Most likely, the remarkable control of tumor growth seen by us and others is due to synergistic effects between NRP1 knockout on both regulatory T cells and CD8+ T cells. As shown in this manuscript, ablation of $Nrp1$ enhances CD8+ T cell migration and effector functions by inhibiting SEMA3A binding. Conversely, research by Vignali and colleagues has shown that NRP1 plays a key role in Treg survival and suppressive capabilities through ligation with Sema4A[62,65], and so affecting NRP1 on both T cell subsets likely allows the immune system to better control tumor growth, than removal of NRP1 on CD8+ T cells alone can. We thus speculate that the effects of SEMA3A on CD8+ T cells are predominant when these effector cells are not inhibited by NRP1+ Treg cells, as indicated here and by Hansen et al.[64], or freed from co-inhibition using anti-PD1 therapy, as shown by Liu et al.[26]. Indeed, when we only affect SEMA3A expression in tumor cells (via over-expression or deletion), tumor growth was not affected, consistent with the hypothesis that atttenuation of Treg's is needed to fully release CD8+ effector T cells.

Why does NRP1 enhance Treg function, but inhibit CD8+ T cells? While not exploring this question in detail, we did find that Treg cells to a larger extent expressed other NRP1 co-receptors, including TGFβRI and II. Indeed, NRP1 has been shown to enhance TGFβ binding in Treg cells[33]. As Treg cells are dependent on TGFβ for their function[66], one intriguing possibility is that NRP1 preferentially partners with these TGFβ-receptors on Treg cells to enable responses to SEMA4A, while CD8+ T cells express Plexin-A1 and Plexin-A4, enabling responses to SEMA3A, which could provide a molecular basis for distinct signaling in each cell type. While we mainly focus on Plexin-A1 in this study, it should be noted that the effects we see from Sema3A on T cells could be explained through both NRP1:Plexin-A1 and NRP1:Plexin-A4 receptor complexes. Plexin-A1 and Plexin-A4 are highly similar both in terms of their intra-cellular signaling domains and both form an almost identical ring-like structure of their extracellular domains[67]. Consistent with this, both proteins have been shown to have identical RapGAP activity[68].

Our study highlights an underappreciated tumor-escape mechanism, namely inhibition of tumor-specific T cells through cytoskeletal paralysis. We find that the effects of SEMA3A on CD8+ T cells are mainly mediated through the co-receptor NRP1, which is retained on recently activated and tumor-specific T cells within the TME. Because SEMA3A mainly binds to these effector and tumor-specific T cells, this inhibitory mechanism is particularly important, and indicate that therapeutic avenues, for example use of antagonistic NRP1 antibodies, could improve anti-tumor immunity. Others have indeed shown that anti-NRP1 antibodies can affect tumor growth in mouse models[19,62,69], likely by both inhibiting Treg cells and, as we show here, by decreasing the effects of SEMA3A on CD8+ effector T cells. Enhancing migration of tumor-specific T cells into tumors is critical for improving the efficacy of checkpoint blockade[70] and adoptive T cell transfer therapies[71], making this an exciting prospect. However, since the SEMA3A-Plexin-A-NRP1 pathway also regulates the maturation of endothelial cells[30] emphasis on timing and drug-target will be critical.

## Methods

### Ethics statement

All animal studies were carried out in accordance with Animals (Scientific Procedures) Act 1986, and the University of Oxford Animal Welfare (AWERB) and Local Ethics Reviews Committee (University of Oxford) under project licence 40/3636.

Acquisition and analysis of ccRCC samples were approved by Oxfordshire Research Ethics Committee C. After informed written consent was obtained, samples were collected and stored until use by Oxford Radcliffe Biobank (project reference 17/A100 and 16/A075). Sex was determined based on sex at birth. Experiments were conducted according to the Declaration of Helsinki conventions.

## Cell lines and media

Cell culture was performed using antiseptic techniques, and grown at 37 °C in a 5% $CO_2$ atmosphere. For some experiments, cells were cultured for 24 h in a 1% $O_2$ chamber. All cell lines were screened for *Mycoplasma*. B16.F10.Ova cell line was generated by transducing B16.F10 with a modified Ovalbumin construct, containing amino-acid 47 – 386 of the full-length ovalbumin, to ensures that Ovalbumin is not secreted. For immunofluorescent studies of B16.F10 tumors in vivo, B16.F10.Ova cells were additionally transduced with an mCherry construct (called B16.F10.Ova.mCherry).

B16.F10.Ova SEMA3A knockout cells were generated using CRISPR/Cas9 genome-editing (see below). B16.F10.Ova SEMA3A over-expressing cells were generated by transducing cells with a lentivirus encoding EFS-*Sema3a* cDNA (NCBI sequence NM_001243072.1)-mCherry, cloned by VectorBuilder (see below).

Adherent cells were kept in DMEM (Gibco, catalog: 11965092), 10% FCS (Gibco, catalog: A5256701), 2 mM L-Glutamine (Gibco, catalog: 25030081), 1 mM Sodium Pyruvate (Gibco, catalog: 11360070), 100 U/mL penicillin + 100 µg/mL streptomycin (Gibco, catalog: 15140122). For some experiments, 10 ng/mL mouse IFNγ (PeproTech, catalog: 315-05) or mouse TNF (PeproTech, catalog: 315-01 A) was added. T cells were kept in "T cell media" consisting of IMDM (Gibco, catalog: 12440053), 10% FCS, 2 mM L-Glutamine, 1 mM Sodium Pyruvate, 1 x Non-essential amino acids (Gibco, catalog: 11140050), 100 U/mL penicillin + 100 µg/mL streptomycin, 10 mM HEPES (Gibco, catalog: 15630080), and 50 µM β-mercaptoethanol (Gibco, catalog: 21985023). PBS (Gibco, catalog: 10010023) was used for washes and as indicated in other experiments.

## Mouse strains and injection of tumor cells, T cells, antibodies and S-Flu H7N1

All experiments were performed in female mice on a C57BL/6 background. Mice were sex-matched and aged between 6 and 12 weeks at the time of the first experimental procedure. CD4-Cre mice were a gift from Katja Simon (NDM, University of Oxford). LifeAct mice were a gift from Shankar Srinivas (DPAG, University of Oxford). C57BL/6, OT-I and CD45.1 mice were purchased from Biomedical services, University of Oxford. *Nrp1*-floxed mice were purchased from Jackson Laboratories (Stock No: 005247). All mice used were on a C57BL/6 background.

Cancer cell lines were split at 1:3 ratio 24 h before injection into mice. On the day of injection, cells were three times in PBS, and $1.5 \times 10^5$ cells in 100 µL PBS were injected intradermally in anesthetized mice. For imaging Sema3A in tumors, $2.5 \times 10^5$ cells were injected in 25% Matrigel Matrix (Corning, catalog: 356234). Six hours prior to tumor harvest, 250 ug brefeldin A (BFA, ChemCruz, catalog: sc-200861A) was injected I.P.

For adoptive transfer of T cells into mice, OT-I splenocytes were stimulated for 48 h using SIINFEKL peptide and sorted as described below, washed two times in PBS and injected via the tail vein.

For infection with S-Flu H7N1, mice were infected intranasally with 10 infectious units S-Flu H7N1 in 50 µL viral growth medium consisting of DMEM with 2 mM Glutamine, 10 mM HEPES, 100 U/mL penicillin + 100 µg/mL streptomycin and 0.1% BSA (VWR, catalog: 1005-70-500 G) under anesthesia.

For CD8-depletion experiments, anti-CD8a (clone 2.43, BioXcell, catalog: BE0061) or IgG2b isotype control (clone LTF-2, BioXcell, catalog: BE0090) were resuspended in PBS and injected intraperitoneally at day -4, -1, 4 and 7 post-injection of cancer cells.

Mice were euthanized by $CO_2$ asphyxiation followed by cervical dissociation.

## Tumor processing and staining for immunofluorescent imaging

For immunofluorescent imaging of immune cells and Sema3A, tumors were immersed, first in paraformaldehyde-based fixative solution (Antigenfix, Diapath, catalog: P0016) overnight and then in sucrose 30% (VWR, catalog: 470302-810) in PBS overnight. Fixed tumors were frozen in Tissue-Tek optimum cutting temperature (OTC) compound (VWR, catalog: 25608-930) and sliced with a cryostat (Leica, CM1900UV) to obtain 10µm cryosections ready to be stained or stored at -80C.

Cryosections were brought to room temperature (RT) and rehydrated in PBS for 10 min, and incubated in blocking solution consisting of 5% donkey serum, 5% goat serum in wash buffer composed of PBS 2% FBS and 0.1% Triton X-100 (Acros Organics, catalog: AC327371000) for a minimum of 4 h at RT. Sections were incubated in the primary antibody mix diluted (see below) in the blocking solution overnight at 4c. Sections were then washed and stained with a secondary antibody mix (see below) for 4 h at RT. Sections were again washed and incubated in DAPI (Sigma, catalog: D9542) at 0.5 µg/ml for 15 min at RT. Slides were mounted with Fluoromount G (SouthernBiotech, catalog: 0100-01) after a final round of washes. Images were taken on the Zeiss LSM980 whole-organ confocal microscope and analysed in Fiji/ImageJ[72].

The following antibodies were used for immunofluorescent imaging: anti-mouse CD8a-AF647 (Abcam, catalog: ab237365, dilution: 1:100), anti-mouse CD31 (BioLegend, catalog: 102501, dilution: 1:100), Goat anti-rat-AF555 (Invitrogen, catalog: A21434, dilution 1:300), Sema3A-AF488 (R&D Systems, catalog: IC1250G, dilution: 1:100).

## Mixed bone marrow chimeras

Mixed bone marrow chimeric mice were made by lethally irradiating male C57BL/6 mice at 4.5 Gy for 300 s twice with 3 h rest between irradiation. Mice were injected i.v. with a 1:1 mixture of CD45.1+ *Cd4*[Cre] *Nrp1*[Flox/+] and CD45.1+CD45.2+ *Nrp1*[Flox/Flox] bone marrow cells. Mice received water containing antibiotics (0.16 mg/mL Enrofloxacin, Bayer Coporation) and were rested for 10 weeks before experimental use.

## Analysis of publicly available transcriptional data

For analysis of SEMA3A co-receptors, we downloaded data collected from the "Immunological Genome Project data Phase 1" (series accession: GSE15907). We focused on CD8+ naïve T cells (accessions: GSM605909, GSM605910, GSM605911) and effector T cells (accessions: GSM538386, GSM538387, GSM538388, GSM538392, GSM538393, GSM538394). Raw expression array files were processed using the affy package[73] and differential expression of selected genes (*CD72, NRP1, NRP2, PLXNA1, PLXNA2, PLXNA3, PLXNA4, PLXNB1, PLXNB2, PLXNB3, PLXNC1, PLXND1, SEMA3A, SEMA3B, SEMA3C, SEMA3D, SEMA3E, SEMA3F, SEMA3G, SEMA4A, SEMA4B, SEMA3C, SEMA4D, SEMA4F, SEMA4G, SEMA5A, SEMA5B, SEMA6A, SEMA6B, SEMA6C, SEMA6D, SEMA7A, TIMD2, HPRT, OAZ1, RPS18, NFATC2, TBX21, EOMES, CD28, PDCD1, CTLA4, LAG3, BTLA, TIM3, ICOS, TNFRSF14, TNFSF14, CD160, CD80, LAIR1, CD244, CXCR1, CXCR2, CXCR3, CXCR4, CXCR5, CCR1, CCR2, CCR3, CCR4, CCR5, CCR5, CCR6, CCR7, CCR8, CCR9, CCR10*) was examined using the limma package[74] in R[75]. Analysis of TCGA data was conducted using TIMER[76].

## Harvesting and activating splenocytes

Following euthanasia and harvest, spleens were strained through a 70 µm nylon mesh (Corning, catalog: 352350) to make a single cell suspension. Cells were washed and resuspended in 3 mL red blood-cell (RBC) lysis buffer (Invitrogen, catalog: 00-4333-57) for 5 min on ice. Cells were washed, counted and resuspended at $2 \times 10^6$ cells per mL in T cell medium. 10 IU/mL IL-2 (PeproTech, catalog: 212-12) and 25 nM SIINFEKL (N4) peptide (Cambridge Bioscience, catalog: SP-MHCI-0016) were added. 200 µL cells were then plated onto a 96-well U-bottom plate and cells allowed to expand for 48 h. For TCR affinity assays, SIINFEKL (N4), SIITFEKL (T4, Cambridge Bioscience, catalog: ANA64403) or SIIQFEKL (Q4, Cambridge Bioscience, catalog: ANA64402) peptides was used at indicated concentrations.

## Sorting CD8⁺ T cells using magnetic beads

CD8a⁺ Negative T Cell Isolation Kit (Miltenyi Biotec, catalog: 130-104-075) was used to sort T cells and was performed according to protocol.

## Preparation of tissue from mice for flow cytometry

When staining cells from B16.F10 and LL/2 tumors, lymph nodes, frontal cortex, lungs or thymus, mice were euthanized, organs harvested and stored in T cell media on ice. Organs were cut into smaller pieces and incubated for 30 min with reagents from a tumor dissociation kit (Miltenyi Biotec, catalog: 130-096-730), strained through a 70 µm nylon mesh, washed and resuspended in 100% Percoll solution (GE Healthcare, catalog: 17-0891-01), and layered carefully on top of 3 mL of 80% and 40% Percoll solution. After 30 min at 2000g, cells at the 80–40% interphase were collected and stained.

## Flow cytometry

Cells were washing and stained in PBS with 2% BSA, 0.1% NaN3 sodium azide (Sigma Aldrich, catalog: S2002). Single color controls were either cells or OneComp Compensation Beads (Thermo Fisher, catalog: 01-1111-41). Viability dyes were used to exclude dead cells from analysis (Zombie Fixable Kit, BioLegend, catalog: 423106, 423114, and 423112). For surface staining, cells were washed and blocked using Fc block (TruStain FcX, clone 93, BioLegend, catalog: 101319, diluted 1:100) for 10 min on ice. Antibody cocktail was added and cells stained on ice for 20 min, in the dark and washed twice. When applicable, cells were fixed in 2% PFA (Sigma Aldrich, catalog: 158127). For quantification of number of cells in organs quantification beads (CountBright, Thermo Fisher, catalog: C36950) were used.

For intracellular staining, cells were fixed in 100 µL/well of FoxP3 IC Perm/fix Buffer according to manufacturer's protocol (Thermo Fisher, catalog: 00-8222-49).

Cells were analyzed on Attune NxT (Life Technologies), LSR Fortessa X20 or X50 (BD Biosciences) flow cytometers in the WIMM Flow Cytometry Facility and data was analysed using FlowJo v10 (FlowJo) and R[75].

The following antibodies and tetramers were used for flow cytometry: CD3e-BV650 (BioLegend, catalog: 344872, dilution: 1:100), CD4-APC710 (Tonbo Bio, Catalog: 20-0041, dilution: 1:200), CD4-BUV810 (BD, catalog: 553730, dilution: 1:100), CD8a-BV711 (BioLegend, catalog: 100748, dilution: 1:100), CD8a- PerCP/ Cy5.5 (BioLegend, catalog: 300924, dilution: 1:100), CD11a-PE (BioLegend, catalog: 101107, dilution: 1:100), CD11b-APC (BioLegend, catalog: 101212, dilution: 1:100), CD19-BV435 (BioLegend, catalog: 115506, 1:100), CD24-APC710 (BD, catalog: 562349, dilution: 1:100), CD25- PerCP/Cy5.5 (BioLegend, catalog: 101912, 1:200), CD31-BV510 (BD, catalog: 563089, dilution: 1:100), CD44-APC/Cy7 (BioLegend, catalog: 103028, dilution: 1:100), CD44-PE/Cy7 (BioLegend, catalog: 103030, dilution: 1:100), CD45-APC (BioLegend, catalog: 304012, dilution: 1:200), CD45.1-FITC (eBioScience, catalog: 11-0453-82, dilution: 1:100), CD45.2- PerCP/Cy5.5 (BioLegend, catalog: 109828, dilution: 1:100), CD49d-BUV395 (BD, catalog: 740219, dilution: 1:100), CD62L-BV610 (BioLegend, catalog: 104408, dilution: 1:100), CD62L-FITC (BioLegend, catalog: 104406, dilution: 1:100), CD69-BUV737 (BD, catalog: 612793, dilution: 1:100), CD103-PE/Cy7 (BioLegend, catalog: 121426, dilution: 1:100), CD105-PE/CF594 (BioLegend, catalog: 562762, dilution: 1:100), CD162-BV421 (BD, catalog: 562807, dilution: 1:100), EpCAM-APC/Cy7 (BioLegend, catalog: 118218, dilution: 1:100), F4/80-BV610 (BioLegend, catalog: 123110, dilution: 1:100), FoxP3-BV421 (BioLegend, catalog: 126419, dilution: 1:50), IFNy-PE (BioLegend, catalog: 505808, dilution: 1:100), Granzyme B-FITC (BioLegend, catalog: 515403, dilution: 1:100), Ly6C-BV780 (BioLegend, catalog: 128016, dilution: 1:100), MHC-II- PerCP/Cy5.5 (BioLegend, catalog: 116416, dilution: 1:100), NRP1-BV421 (BioLegend, catalog: 145209, dilution: 1:100), NRP1-PE (BioLegend, catalog: 145204, dilution: 1:100), NRP1-BV421 (BioLegend, catalog: 354514, dilution: 1:100), HLA-A2/ SLLMWITQV APC (In-house generated, dilution: 1:100), H-2Dᴮ-

NP PE/Cy7 (In-house generated, dilution: 1:100), H2-Kb/SIINFEKL APC (In-house generated, dilution: 1:100), PD-1- eFluor610 (eBioScience, catalog: 61-2799-42, dilution: 1:100), PD-1-APC (BioLegend, catalog: 329908, dilution: 1:100), Plexin A1-PE (R&D Systems, catalog: FAB4309P, dilution: 1:50), Plexin A2- APC (R&D Systems, catalog: FAB5486A, dilution: 1:50), Plexin A4-PE (Abcam, catalog: ab39350, dilution: 1:100), Podoplanin-APC (BioLegend, catalog: 127410, dilution: 1:100), TCRab-FITC (BioLegend, catalog: 306706, dilution: 1:100), TCRb-APC/Cy7 (BioLegend, catalog: 109220, dilution: 1:100), TCRb-PE-CF594 (BD, catalog: 562841, dilution: 1:100), TGFbRI-PE (R&D Systems, catalog: FAB5871P, dilution: 1:50), TGFbRII (R&D Systems, catalog: FAB532P, dilution: 1:50), TNFa-PerCP/Cy5.5 (BioLegend, catalog: 506322, dilution: 1:100), Sema3A-AF488 (R&D Systems, catalog: IC1250G, dilution: 1:100), Sema3A-PE (R&D Systems, catalog: IC1250P, dilution: 1:100), VEGFR2-PE (BioLegend, catalog: 136404, dilution: 1:100).

## RT-qPCR

RNA was extracted from cells using RNeasy kit (QIAGEN, catalog: 74104), followed by quantification on Nanodrop (Thermo Scientific). RNA was reverse transcribed using QuantiTect Reverse Transcription Kit (QIAGEN, catalog: 205311). Controls without RNA or reverse transcription were included. All experiments were performed in technical triplicates and biological duplicates. cDNA was diluted to 10-20 ng in 5 µL/well and added to qRT-PCR plates. TaqMan probes were combined with 2x Fast TaqMan Master Mix and 5 µL/well added to the cDNA. qRT-PCRs were run on a QuantStudio7 qRT-PCR machine (Life Technologies). Expression was normalized to the house keeping gene *Hprt* (ThermoFisher Scientific, TagMan Probe, catalog: Mm03024075_m1). The following TaqMan probes (all from Thermo-Fisher Scientific) were used: *Bnip3* (catalog: Mm01275600-g1), *Hprt* (catalog: Mm03024075-m1), *Pdk1* (catalog: Mm00554300-m1), *Pdl1* (catalog: Mm00452054-m1), *Sema3a* (catalog: Mm00436469-m1) and *Vegfa* (catalog: Mm00437306-m1).

## Western blot

Cells were washed in PBS and pelleted, before being resuspended in lysis buffer with a protease inhibitor cocktail (Roche, catalog: 4693159001). Cell-debris was removed by centrifugation at 4 °C. Supernatant was collected and quantified using BCA protein assay (Pierce BCA Protein Assay Kits, Thermo Scientific, catalog: 23225).

Samples were mixed with LDS Sample Buffer (Invitrogen, catalog: NP0007) and Sample Reducing Agent (Invitrogen, catalog: NP0004) and heated to 95 °C for 5 min. 4–12% Bis-Tris gels (Invitrogen, catalog: NP0321BOX) and MES SDS Running Buffer (Invitrogen, catalog: NP0002) were used for proteins with a molecular weight below 200 kDa, while proteins above 200 kDa were blotted on a 3–8% Tris-Acetate gels (Invitrogen, catalog: EA0375BOX) in MOPS Running Buffer (Invitrogen, catalog: NP0001). Proteins were separated at 200 V for -1 h and transferred using the TransBlot Turbo Transfer (BioRad) system. Gels were blocked in 5% BSA/PBS solution (blocking buffer) for at least 30 min at RT. Membranes were stained with primary antibody in fresh blocking buffer and incubated at 4 C overnight on a shaker, washed five times in PBS with Tween-20 (Thermo Scientific, catalog: 85113) followed by incubation with fluorescent secondary antibodies (VRDye, LI-COR, catalog: 926-49010, 1:20.000) in blocking buffer for 1 h on a shaker. Membranes were imaged using the Odyssey Near-Infrared imaging system (LI-COR). The following antibodies were used: Anti-NRP1 antibody (Abcam, ab184783), anti-PlexinA1 (R&D Systems, AF4309), anti-PlexinA2 (R&D Systems, AF5486), anti-GAPDH (Santa Cruz, sc-32233), anti-β-Actin (Cell Signaling Technology, 13E5).

## CRISPR/Cas9 editing and verification of B16.F10.Ova cells

A sgRNA targeting *Sema3a* was cloned into Cas9-2A-EGFP expression vector pX458. This vector was electroporated into $1 \times 10^6$ B16.F10.Ova

cells suspended in Solution V (Lonza, catalog: VCA-1003) using the Amaxa 2B nucleofector (Lonza) with settings P-020. Thus, both Cas9 and sgRNA was only transiently expressed in the cells, to avoid unintended immunogenicity of the bacterial protein. After 48 h, single cells were sorted using the SH800 cell sorter (SONY) and expanded. Clones were genotyped by high-throughput sequencing. Briefly, the targeted locus was PCR amplified from each clone and subsequently indexed with a unique combination of i5 and i7 adapter sequences. Indexed amplicons were sequenced on the MiSeqV2 (Illumina, catalog: MS-102-2001) and demultiplexed reads from each clone were compared to the wild-type *Sema3a* reference sequence using the CRISPResso webtool[77].

### Lentiviral transduction of cells

A lentiviral *Sema3A* overexpression vector containing mCherry was purchased from VectorBuilder. To generate viral particles, this transfer vector was co-tranfected into HEK-293T cells along with the packaging and envelope plasmids pCMV-dR8.91 and pMD2.G. Viral supernatant was filtered and used to transduced B16.F10.Ova cells. mCherry positive cells were selected by FACS using the SH800 cell sorter (SONY).

### Protein production of SEMA3A and mutant SEMA3A

Recombinant mouse SEMA3A$_{S-P}$ (residues 21–568 plus a C-terminal 6-His tag), SEMA3A$_{S-P-I}$ (residues 21–675, with the 6-HIS tag removed after purification) along with the NRP1-binding deficient mutant SEMA3A (residues 21–568, L353N-P355S), here called mutant SEMA3A, were cloned into a pHLsec vector as described[78]. The L353N P355S mutation in mutant SEMA3A introduces an N-linked glycan to the SEMA3A-NRP1 interaction site, which blocks the formation of SEMA3A-NRP1-PLXNA2 signaling complex as described[16]. In all proteins, furin sites (R551A and R555A) were removed to prevent SEMA3A proteolytic processing. Proteins were expressed in HEK293T cells and purified from buffer-exchanged medium by immobilized metal-affinity followed by size-exclusion chromatography using a Superdex 200 column (GE, catalog: 28990944).

For some experiments, purified SEMA3A protein was labeled with AF647 using Alexa Fluor 647 Antibody Labeling Kit according to protocol (Invitrogen, catalog: A20186) at a F/P rate at 1–2.

### Live-cell imaging of cells for migration studies, LifeAct and IRM quantification

μ-Slide 8 well (Ibidi, Cat. no. 80826) and μ-Slide Angiogenesis (Ibidi, Cat. no. 81506,) were used for live-cell imaging of T cells.

Proteins in PBS were adsorbed to tissue culture plates for 2 h as RT, followed by three washes in PBS with 1–2% BSA. The following proteins and concentrations were used: ICAM-1-Fc (BioLegend, catalog: 553006, at 10 μg/mL), anti-CD3 (BioLegend, catalog: 145-2C11 at 10 μg/mL), CXCL12 (BioLegend, catalog: 578706 at 0.4 μg/mL, BioLegend). Sema3A$_{S-P}$ and mutant Sema3A were either adsorbed to surfaces in a similar manner or added to the medium while imaging at 5 μM. Plates were used immediately after preparation.

For experiments, T cells were activated for 48 h, sorted and washed. In cases were αNRP1 or isotype control treatment was applied, T cells were incubated with these for 15 min at 37 °C and washed before use. T cell medium without phenol red was used. Cells were added to plates placed on a stage in environmental chamber at 37 °C at the microscope. DeltaVision Elite Live cell imaging microscope, Zeiss LSM 780 or 880 confocal microscopes were used with Zeiss Plan-Neofluar 10 x (0.3 NA), 40 x (0.6 NA) or 63 x (1.3 NA) lenses.

In assays with flow, T cells were injected into the to the μ-Slide I 0.4 Luer (Ibidi, catalog: 80172) and allowed to adhere for 10 min at 37ºC prior to initiation of flow with a Harvard Apparatus PHD 2000 pump connected through a tube adapter set (Ibidi, catalog: 10831). Near wall flow velocities were measured using fluorescent beads.

In cases were cells were treated with drugs while imaging, drugs were first resuspended in T cell medium and carefully added on top of the solution while images were being acquired. The following drugs were used: Jasplakinolide (catalog: J4580, Sigma, used at 5 μM) and Blebbistatin (catalog: 203390, EMD Millipore, used at 100 μM), as well as Sema3A$_{S-P}$ (used at 1–5 μM).

Data were acquired at 1 s to 1 min per frame as indicated and analyzed in Fiji/ImageJ[72]. For cell tracking the Trackmate package was used[79]. Tracks were analyzed in R[75] and visualized using the ggplot2 package[80]. For cell contours, IRM and LifeAct data were thresholded and collected in Fiji/ImageJ, then exported to R for analysis and visualization. Quantification of IRM and LifeAct area while adding drugs, movies were edited such that the analyzed frames were equal to the timing indicated in figures. In videos with a frame-rate of 1 frame per second, 3 frames around the frame in which drugs were added was removed to avoid blurry images.

### Transwell chemotaxis assay

Trans-migration was assessed in 24-well transwell plates with 3 μm pore size (Corning Life Sciences, catalog: 3472). The lower chamber was loaded with 500 μL T cell medium with 10 IU IL-2 and with or without 50 ng/mL CXCL10. $1 \times 10^5$ OT-I CD8$^+$ T cells were stimulated and sorted and added in a volume of 100 μL of T cell medium to the upper chamber, in either the presence of a blocking αNRP1 antibody (R&D Sytems, catalog AF566, 5 ug/mL) or an isotype control and 5 μM Sema3A$_{S-P}$. As a positive control, effector cells were placed directly into the lower chamber. As a negative control, migration medium alone was placed in the upper chamber. Plates were incubated for 3 h at 37 °C in a 5% $CO_2$ atmosphere. Cells from the lower chamber were quantified by flow cytometry.

### Immunological synapses analysis on supported lipid bilayers (SLB)

The concentrations of the ligands used were as follows: 5 μg/mL to achieve 10 molecules/μm$^2$ of biotinylated H2Kb-SIINFEKL, 68 ng/mL to achieve 100 molecules/μm$^2$ of 12x-HIS-tagged CD80 (AF488-labeled), and 122 ng/mL to achieve 200 molecules/μm$^2$ of 12x-His tagged ICAM-1 (AF405-labeled). These concentrations were determined based on titrations on bead supported bilayers analyzed by flow cytometry. SEMA3A$_{S-P-I}$ (described above) was used as this protein had no HIS-tag and so could not interfere with binding of other tagged proteins in bilayer.

For live cell imaging, supported lipid bilayer presenting H2Kb-SIINFEKL, CD80, and ICAM-1 were assembled in sticky-Slide VI 0.4 Luer (Ibidi, catalog: 80608) channels. The entire channel was filled with a liposome suspension to form a bilayer all along the channel. Live cells on SLBs were imaged using the Olympus FluoView FV1200 confocal microscope that was enclosed in an environment chamber at 37 °C and operated under standard settings. x60 oil immersion objective (1.4 NA) was used with x2 digital zoom for time-lapse imaging at 20 s intervals.

For fixed cell imaging, SLBs presenting H2Kb-SIINFEKL, CD80, and ICAM-1 were assembled in 96-well glass-bottom plates (catalog: MGB096-1-2-LG-L, Brooks). $5 \times 10^4$ cells were introduced into the wells at 37 °C and fixed 10 min later by adding 8% PFA in 2x PHEM buffer. After 3 washes with 0.1% BSA in HBSS (Gibco, catalog: 14025092), the fixed cells were imaged on the InCell 6000 microscope using a 40x air objective (0.75 NA) at 16 locations per well.

Analysis of fixed cell images was carried out by the MATALB based TIAM HT package[41]. The source-code is available at https://doi.org/10.5281/zenodo.10522993.

### Analysis of CDR3 and TRBV usage in CD8$^+$ ccRCC TILs

cDNA from single cells was obtained using a modified version of the SmartSeq2 protocol[81]. Briefly, single cells were sorted into plates

containing lysis buffer and cDNA was generated by template switch reverse transcription using SMARTScribe Reverve Transcriptase (Clontech, catalog: 639538), a template-switch oligo and primers kit designed for the constant regions of Trac and Trbc genes. TCR amplification was achieved by performing two rounds of nested PCR using Phusion High-Fidelity PCR Master Mix (New England Biolabs, catalog: M0531S). During the first PCR priming, indexes were included to identify each cell. A final PCR was performed to add the Illumina adapters. TCR libraries were sequenced on Illumina MiSeq using MiSeq Reagent Kit V2 300-cycle (Illumina). FASTQ files were demultiplexed for each cell. Sequences from clones were analysed using MiXCR[82]. Post-analysis was performed using VDJtools[83]. The CDR3 and TRBV usage data generated in this study have been deposited in the Sequence Read Archive under accession code PRJNA1075074.

### CT-tetramer staining

HLA-A2 monomers were made in-house, and loaded with CT-antigens through UV-directed ligand exchange using published protocols[84]. Following ligand exchange, all monomers were tetramerized through binding to PE-Streptavidin, washed, and combined to allow for 'cocktail' staining of cells. Frozen vials of tumor tissue were thawed, dissociated, and CD45-positive sorted, followed by staining with 0.5 μg of each tetramer in 50 μL for 1 h at RT. Otherwise staining proceeded like described previously.

The following CT-antigens (ligands) were loaded into HLA-A2 monomers: LAGE-1 (MLMAQEALAFL), LAGE-1 (SLLMWITQC), NY-ESO-1 (SLLMWITQA), MAGE-A1 (KVLEYVIKV), MAGE-A10 (GLYDGMEHL), MAGE-A2 (YLQLVFGIEV), MAGE-A3 (FLWGPRALV, KVAELVHFL or FLWGPRALV), MAGE-C2 (ALKDVEERV, KVLEFLAKL or LLFGLALIEV), MART1 (ELAGIGILTV), Meloe-1 (TLNDECWPA), PRDX5/OMT3-12 (AMAPIKVRL), RAGE-1 (LKLSGVVRL), SSX2 (KASEKIFYV), TRP2 (SVYDFFVWL), Tyrosinase (YMDGTMSQV), BING4 (CQWGRLWQL), GnT-V (VLPDVFIRCV).

### Immunohistochemistry and image acquisition and analysis

Diagnostic hematoxylin and eosin (H&E) stained slides for cases of ccRCC were reviewed to identify corresponding formalin-fixed paraffin embedded tissue blocks that contained both tumor and adjacent non-tumor tissue. Strictly serial 4 μm sections were cut from the most appropriate block. These sections underwent immunohistochemistry staining on a Leica BOND-MAX automated staining machine (Leica Biosystems). Briefly, sections were deparaffinized, underwent epitope retrieval and endogenous peroxidase activity was blocked with 3% hydrogen peroxide (5 min). Subsequently, sections were incubated with the primary antibody (30 min) followed by post-primary and polymer reagents (8 min each). Next, 3,3'-Diaminobenzidine (DAB) chromogen was applied (10 min) (all reagents contained within the BOND Polymer Refine Detection kit, Leica Biosystems, catalog: DS9800). For double immunohistochemistry staining, the above cycle was repeated twice with the first cycle using Fast red chromogen labeling and a second cycle with DAB chromogen labeling. At the end of both the single and double immunohistochemistry protocols, the sections were counterstained with hematoxylin (5 min), mounted and left to dry overnight. The following primary antibodies were used during staining: CD31 (Agilent Technologies, catalog: JC70A, dilution: 1:800), CD8 (Agilent Technologies, catalog: C8/144B, dilution: 1:100) and SEMA3A (Abcam, catalog: ab199475, dilution: 1:4000).

Stained slides were scanned at x400 magnification using the NanoZoomer S210 digital slide scanner (Hamamatsu). SEMA3A-stained digital images were reviewed by a trained pathologist (PSM) and the extent of staining was quantified in the regions where expression was deemed to be highest ('SEMA3A rich') and lowest ('SEMA3A poor') within the same tumor, using custom-made MATLAB (MathWorks) scripts (% staining = DAB$^+$ pixels/total pixels x 100). Image analysis were performed by a trained histopathologist (P.S.M.) who reviewed the images and selected the area with most and least CD31 staining in a blinded fashion to the SEMA3A and CD8 slides. The extent of staining was quantified in the regions where expression was deemed to be highest and lowest (% stained area = positively stained pixels/total pixels x 100). This analysis was repeated in the same regions on adjacent serial sections for SEMA3A (% stained area = positively stained pixels/total pixels x 100) and CD8 (discrete cell counts were calculated from positively stained regions using a watershedding process). Analysis scripts used here have been described in Bull JA et al.[85] and are available at https://doi.org/10.5281/zenodo.10625167.

### Statistics and schematics

Statistical analysis was performed in Prism software (GraphPad) or R[75]. Data was tested for Gaussian distribution. For multiple comparisons, either one-way or two-way analysis of variance (ANOVA) was used with Tukey's test to correct for multiple comparisons. For comparison between two groups, Student's t-test or one-tailed or two-tailed Mann–Whitney test were used. Image schematics were either designed by the authors (Fig. 2f, g and Fig. 3c) or were created with BioRender.com (Fig. 6b).

### Reporting summary

Further information on research design is available in the Nature Portfolio Reporting Summary linked to this article.

## Data availability

The CDR3 and TRBV usage data generated in this study have been deposited in the Sequence Read Archive under accession code PRJNA1075074. Analysis of survival data in ccRCC patients was done on TCGA data available at the TIMER website: http://timer.cistrome.org/. Analysis of SEMA3A receptors was done on data from "Immunological Genome Project data Phase 1" (series accession: GSE15907). All other data are available in the article and its Supplementary files or from the corresponding author upon request. Source data are provided with this paper.

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

## Acknowledgements

The authors wish to thank members of the Vincenzo Cerundolo and Tudor A. Fulga laboratories, University of Oxford, for helpful discussions and suggestions. Simon Davis and Oliver Bannard, University of Oxford, for helpful advice and guidance. The staff of the University of Oxford Department of Biomedical Services for animal husbandry. Christoffer Lagerholm and Dominic Waithe of the Wolfson Imaging Centre at University of Oxford for microscopy training and support. Helena Coker and Kseniya Korobchevskaya from the Oxford-ZEISS Centre of Excellence in Biomedical Imaging at Kennedy Institute of Rheumatology which is supported by the Kennedy Trust for Rheumatology Research, IDRM and Carl Zeiss GMBH. The Medical Research Council (MRC) WIMM Flow Cytometry Facility for training and support. The Oxford Centre for Histopathology Research and the Oxford Radcliffe Biobank, which are supported by the NIHR Oxford Biomedical Research Centre. Lisa Browning, Oxford University Hospital, for examination of histology samples. David Pinto, University of Oxford, for code for analysis of CDR3 sequences. This work was supported by the U.K. MRC (MRC Human Immunology Unit), the Oxford Biomedical Research Centre, and Cancer Research UK (CRUK) through CRUK Cancer Centre (C399/A2291 to V.C.; C38302/A17319 to V.K.W.; C375/A17721 to E.Y.J.; 29549 to A.G.; CTRQQR-2021\100002 to J.A.B.); the Wellcome Trust (212343/Z/18/Z to M.F.; 100262Z/12/Z to M.L.D. and S.V.) and Kennedy Trust for Rheumatology Research (to M.F., M.L.D., S.V., A.G., J.M.M.), European Commission (ERC-2014-AdG_670930 to M.L.D. and V.M.), Cancer Research Institute (to V.M.), and EPSRC (EP/S004459/1 to M.F. and H.C.Y.). The Wellcome Centre for Human Genetics is supported by Wellcome Trust Centre grant 203141/Z/16/Z. P.S.M. is supported by a Jean Shanks Foundation/Pathological Society of Great Britain & Ireland Clinical Research Training Fellowship. C.K. is supported by a Wellcome Studentship (105401/Z/14/Z). V.J. is supported by an EMBO Long-Term Fellowship (ALTF 1061–2017). J.A.B. is supported by EPSRC/MRC Centre for Doctoral Training in Systems Approaches to Biomedical Science (EP/G037280/1) and the EPSRC Impact Acceleration Account (EP/R511742/1). L.R.O. is supported by the Independent Research Fund Denmark (8048-00078B). V.K.W. is supported by CRUK Oxford Centre Prize DPhil Studentship. M.B.B. is supported by Early-Career Clinician Scientists fellowship from the Lundbeck Foundation (R381-2021-1278). A.V.H. was supported by a Wellcome Trust Clinical Research Fellowship (106287/Z/14/Z) and an A.G. Leventis Foundation Scholarship. A.V.H. is currently supported by a NIHR/University of Cambridge Clinical Lectureship

(RC30016) and a Clinical Lecturer Starter Grant from the Academy Of Medical Sciences (G122195; RDAG/600).

## Author contributions

*Conceptualization*: M.B.B, V.C., E.Y.J., M.L.D., T.A.F., M.F., A.G.; *Formal analysis*: M.B.B., Y.S.M., V.A., M.R., P.S.M., V.M., M.F., A.G.; *Funding acquisition*: M.B.B., V.C., E.Y.J., M.L.D., T.A.F., M.F., C.W.P.; *Investigation*: M.B.B., Y.S.M., V.A., P.S.M., U.G., S.V., M.R., C.K., J.C., A.V.H., V.M., P.R., L.R.O., M.F., H.C.Y., J.M.M, G.M.; *Methodology*: V.C., U.G., M.B.B., M.L.D., M.F., H.C.Y., Y.S.M., A.R.T., A.G.; *Project administration*: V.C., M.B.B.; *Resources*: U.G., S.V., V.J., M.R., V.K.W., Y.K., V.M., A.R.T., A.G., M.B.B.; *Software*: M.R., J.A.B., M.B.B., V.M.; *Supervision*: V.C., E.Y.J, M.L.D, A.G.; *Visualization*: M.B.B., P.S.M., V.M., M.F.; *Writing – original draft*: M.B.B., V.C., E.Y.J., M.L.D., M.F.; *Writing – review & editing*: M.B.B., V.C., E.Y.J., M.L.D., M.F., Y.S.M., C.K., P.S.M., V.J.

## Competing interests

The authors declare the following competing interests: M.B.B. has received consulting honorariums from Janssen, Roche and Kite/Gilead, unrelated to the present work. Y.S.M. has consulted for Apiary Therapeutics, Notch Therapeutics and CCRM, unrelated to the present work. The remaining authors declare no competing interests.

## Additional information

[1]MRC Human Immunology Unit, MRC Weatherall Institute of Molecular Medicine, University of Oxford, Headley Way, Oxford OX3 9DS, UK. [2]MRC Weatherall Institute of Molecular Medicine, University of Oxford, Headley Way, Oxford OX3 9DS, UK. [3]Nuffield Department of Medicine, University of Oxford, Nuffield Department of Medicine Research Building, Roosevelt Drive, Oxford OX3 7FZ, UK. [4]Kennedy Institute of Rheumatology, University of Oxford, Roosevelt Dr, Oxford OX3 7FY, UK. [5]Division of Structural Biology, Wellcome Centre for Human Genetics, University of Oxford, Roosevelt Drive, Oxford OX3 7BN, UK. [6]Wolfson Centre for Mathematical Biology, Mathematical Institute, University of Oxford, Radcliffe Observatory Quarter, Woodstock Road, Oxford OX2 6GG, UK. [7]Department of Health Technology, Technical University of Denmark, Ørsteds Plads, Building 345C, 2800 Kgs Lyngby, Denmark. [8]Present address: Centre for Cellular Immunotherapy of Haematological Cancer Odense (CITCO), Department of Clinical Immunology, Odense University Hospital, University of Southern Denmark, Odense, Denmark. [9]Present address: Paul Albrechtsen Research Institute, CancerCare Manitoba, 675 Mcdermot Ave, Winnipeg, MB R3E 0V9, Canada. [10]Present address: Department of Biochemistry and Medical Genetics, Rady Faculty of Health Sciences, University of Manitoba, Bannatyne Ave, Winnipeg, MB R3E 3N4, Canada. [11]Present address: Instituto de Medicina Molecular João Lobo Antunes, Faculdade de Medicina, Universidade de Lisboa, Lisbon, Portugal. [12]Present address: Pulmonary and Critical Care Medicine, Oregon Health and Science University, Portland, OR, US. [13]Present address: Division of Gastroenterology & Hepatology, Department of Medicine, Cambridge University Hospitals, University of Cambridge, Cambridge, England. [14]Present address: Early Cancer Institute, Department of Oncology, University of Cambridge, Cambridge, England. [15]Deceased author: Vincenzo Cerundolo. ✉e-mail: mike.bogetofte.barnkob@rsyd.dk; yvonne@strubi.ox.ac.uk

