## [Peer Review File · Nature Communications]

Semaphorin 3A causes immune suppression by inducing cytoskeletal paralysis in tumour-specific CD8+ T cellsEditorial Note: Parts of this Peer Review File have been redacted as indicated to remove third-party material where no permission to publish could be obtained.

REVIEWER COMMENTS

Reviewer #1 (Remarks to the Author):

In this manuscript, the authors address a contentious issue as to whether SEMA3A is pro-tumorigenic or anti-tumorigenic. They present data suggesting that SEMA3A acts through NRP1 on T cells to increase tumorigenicity, and reduce immunogenicity, by freezing or trapping activated anti tumor T cells in areas of high SEMA-3A expression in the tumor or in the tumor associated vasculature. They correlate extensive murine studies with some patient data showing that in human patients with clear cell renal carcinomas NRP1 T cells were trapped in SEMA3-A rich areas and SEMA3-A expression correlated with reduced survival.

Specific comments:

The authors have provided some elegant studies, principally (but not exclusively) with OT-I T cells that NRP1 is upregulated on T cells when they encounter antigen in an antigen concentration and TCR recognition dependent fashion. Their data shows that NRP-1 deficient mice controlled B16 tumors better than NRP-1 intact mice suggesting that NRP-1 in T cells was causing some sort of inhibition to T cell control. Increased tumor control was associated with increased levels of CD8+ T cells in the tumors. They also knocked out SEMA-3A or overexpressed it in B16ova cells.

The authors then show that the OE and KO B16-ova tumors grow at the same rates in mice. If I am understanding this correctly this seems like a disconnect to me with Fig.2A/B where B16 tumors grow much less well in NRP1 KO mice implying T cell mediated control of these tumors. If Nrp1 KO mice control B16 tumors better than Nrp1 intact mice – dependent upon CD8+ T cells - that implies that B16 growth in these mice is CD8 mediated. In that case would you not expect that the B16-ova tumors with SEMA-3A KO would be controlled much

more effectively than OE tumors, especially since the ova tumors have an even better ag to stimulate anti-tumor T cells with? Is this a disconnect between Figs 2A,B and 2G?

In this respect much of the pre-clinical data is performed with OT-I T cells. Related to the points above, if B16 tumors are well controlled in NRP-1 KO mice, then it should be possible to show these effects also for truly endogenous anti-tumor T cells. For example, in mice vaccinated against B16 tumors (eg with B16-GM-CSF+anti-CTLA4) endogenous anti-tumor T cell responses will be generated. Under these circumstances how would SEMA-3A KO or OE tumors fare in regular mice? Would the endogenous anti B16 T cell responses control SEMA-3A KO tumors much better than regular B16 tumors or OE B16 tumors? Such data would add confidence that the effects seen with OT-I T cells were also relevant to endogenous anti-tumor T cells recognizing 'real' tumor antigens.

With KO and OE tumors on different flanks + act of OT-I the KO tumors grew significantly more slowly than the OE tumors with more infiltration of OT-1.

Showed that SEMA-3A significantly affects T cell adhesion and mobility and prevents formation of the immunological synapse by confocal.

The authors also show that SEMA-3A is associated with poor survival in clear cell renal cell carcinoma patients. Is this association also true in other cancers? Especially those amenable to ICB where presumably anti tumor T cells are available and can be inhibited if SEMA-3A is expressed? One would expect that such an association would be widespread if SEMA-3A expression is critical to T cell control of tumors then there should be a clear association in immunogenic tumors which respond to immune checkpoint blockade and a negative association in those that do not.

In patient samples SEMA-3A co-localized with the endothelial marker CD31 and CD8 T cells clustered in the regions of SEMA-3A expression. The authors suggest that this shows that SEMA-3A expression is associated with tumor vasculature and that these areas trap CD8+ T cell in the tumor. If that is true, can they confirm that in their pre-clinical models? Thus, in a

B16ova tumor that is seeded with, for example, 20% SEMA-3A OE cells and 80% KO cells one would predict that infiltration of adoptively transferred OT-I T cells would cluster around SEMA-3A OE cells and be reduced elsewhere in these tumors compared to say a 100% KO tumor; similarly the control of a 20% OE/80% KO tumor by OT-I T cells would then be expected to be worse than a 100% KO tumor with OT-I clustered around the islands of SEMA-3A positivity. These sorts of expts would be very valuable to refute the possible explanation of the human histology that SEMA-3A expression favors T cell infiltration into tumors (because that is where the T cells are seen) and that a lack of SEMA-3A expression is actually contributing to the immunological desert -which is counter to the authors' hypothesis.

Have the authors tried to use their data for therapy –showing that NRP-1 is an important checkpoint by using an antagonistic NRP-1 antibody in vivo – that is different from using the KO and OE cells used here which cannot provide a therapeutic outlet for this work in the clinic.

In summary, this is an extensive manuscript with well controlled and well performed experiments. The data show that SEMA-3A may play an important role in limiting T cell mediate tumor immunity and the experiments shown are largely convincing. In my opinion, the conclusions could be significantly strengthened by generalizing the results away from just the use of the OT-I system; by extending the importance of the correlative data from ccRCC to other human tumors, by showing that what is observed in human tumors holds true in the pre-clinical models in terms of the association of T cells with SEMA-3A^{hi} regions of tumors to show that SEMA-3A is truly inhibitory rather than stimulatory for T cell infiltration and by showing how these data could be used clinically in a pre-clinical model (ie with an anti-NRP-1 Ab type strategy). That said, there is already a large amount of data included and the authors may be able to address some of these concerns without necessarily demanding extensive new data.

Reviewer #2 (Remarks to the Author):

The manuscript written by Dr. Mike B Barnkob, et al. reveals novel function of SEMA3A in

TME. SEMA3A acts as an inhibitor of tumor associated CD8+ T cells via its receptor Nrp1, leading to cellular paralysis. Their theory was also confirmed by examining a clear cell renal cell carcinoma patient, found that SEMA3A expression is associated with reduced survival and CD8+T cells accumulation. The manuscript is well structured and substantial experiments are conducted. Most of them seem reasonable and I basically feel positive for publication of this work. However, some important points should be addressed and clarified. Also, there are many careless mistakes in figures. I recommend authors should review the details of manuscript carefully and revise it to correct any tiny mistakes (As far as I noticed, I described in the section <minor remarks>).

<Major remarks>

1) In Fig. 1G and Sup Fig. 1D, authors identify the main receptor as PlexinA1 because of its high expression on CD8+T cells. However, as described in Line 120, PlexinA4 is also expressed on CD8+T cells. Previous reports have shown that SEMA3A-PlexinA4, unlike PlexinA1, activates macrophage inflammation (JEM 2010). As such, PlexinA4 may send signals in a different direction than PlexinA1. If authors think PlexinA4 is not involved in the theories of this paper, please prove it (e.g. by using knockdown or knockout cell lines), or at least it should be discussed.

2) In Fig. 2 and Sup Fig. 3, results between flu infection model and tumor model are different. Authors explained that tumors and endothelial cells strongly express SEMA3A in hypoxic TME. Regarding tumor cells, it is reasonable because SEMA3A expression level is up-regulated under hypoxia and SEMA3A-KO tumors grow slower. Regarding endothelial cells, percentage of SEMA3A positive cells in tumor LEC/BEC (Sup Fig.2E) is larger than that of infected lung (Sup Fig.2B). However, whole lung LECs and BECs in pneumonia and LECs and BECs in the tumor environment may be essentially different.

2-1) Do authors simply see the difference of SEMA3A expression between mature endothelial cells that are originally present in lung and neoplastic endothelial cells? Or do authors think SEMA3A expression in LECs/BECs is also induced by hypoxia? Please describe in the manuscript.

2-2)

-if yes for former question in 2-1)-

In pneumonia, there are some localized hypoxic environments at inflammatory foci. Show IHC or other convinced data looking at local inflammatory foci to explain even in hypoxic-estimated environment, SEMA3A expression is still low and CD8+T accumulation is normal in the lung.

- if yes for latter question in 2-1)-

Show the data that the elevation of SEMA3A in endothelial cells is induced by hypoxia and discuss why it does not happen inflammatory hypoxia foci in pneumonia.

2-3) what is the molecular signal mechanism under hypoxia condition that can induce SEMA3A expression in tumor and/or other cells?

These questions need to be addressed to clarify whether the theory in this paper is truly tumor-specific or not.

3) SEMA3A is produced by not only tumors and endothelial cells. For example, CD16+ monocyte is one popular source of SEMA3A in inflammatory condition. I understand Fig.2 C-E clearly show the effect is mainly involved in CD8+T cells, but authors should show the profiles of SEMA3A expression in other key players in TME (either or both human/mice TME) to increase the generality of this work.

4) In Fig6 K-M, how do authors separate the SEMA3A Rich from the Poor? Authors describe they construct mathematical scripts; This would be not a popular method. I'd like to request clear and brief explanation how to do it, with visual examples. I mean, I want to know whether this method is sufficiently satisfactory for general readers.

<Minor Remarks>

Fig Sup 1a: Briefly describe the method how to do the data mining from public database.

Fig 1b: Vertical axis numbers are missing.

Fig 1d: Which is naïve, which is stimulated?

Fig 2c: Blue and red graph should mean Flox/Flox. "ns" is missing.

Fig 2d: Blue dot should mean Flox/Flox.

Fig 2G: Horizontal axis numbers are missing.

Line 183: not 4M, it should be 2M.

Fig 3H: Vertical axis numbers are missing. Also, some reagents in horizontal bar are missing.

CXCL10, SEMA3A, what and what?

Fig 4B: Vertical axis numbers are partially missing.

Fig 6FIJ: Vertical axis numbers are partially missing.

Fig 6DF: clarify blue dots mean tumor adjacent tissue.

Reviewer #3 (Remarks to the Author):

The current article has attempted to establish a SEMA3A-NRP1 immune checkpoint on CD8 T cells by interrupting F-Actin and CD8 T cell migration in the tumor microenvironment. There is no doubt that the find could be potentially important due to the novel link from cancer cell-produced SEMA3A to suppress NRP1+ CD8 T cells by inactivating F-Actin and immobilization of CD8 T cells. The article at its current format, however, has some major flaws.

Major critiques:

1) No explanation of discrepancy with some very critical publications in the NRP1 field related to Tregs and CD8 T cells, which leads to the potentially wrong interpretation of tumor immune phenotype from CD4-cre/NRP1fl/fl mice published by the Helfrich group (PMID: 23045606). The published JEM paper used the same model and showed that Nrp1+ Tregs are the most critical cell types for tumor promotion, leading to the suppression of CD8 T cells. Specific deletion of NRP1 in CD8 T cells (E8I cre, only deplete NRP1 in CD8 cells, PMID: 32661362) has no phenotype in primary tumor growth (B16 melanoma, the same model used in this paper). This further supports the primary tumor phenotype in CD4-cre/NRP1fl/fl mice is due to Tregs. Further clarification is required since this will directly impact the significance of this study: whether or not SEMA3A has its impact on CD8 T cell via NRP1.

2) The SEMA3A and NRP1 connection on CD8 T cells is not fully established. In addition to

the concerns raised in 1), the functional connection between SEMA3A and NRP1 within CD8 T cells are not fully established. Figs3-5 aim to study the function of SEMA3A in CD8 T cells. NRP1 KO CD8 T cells should be used and compared to draw a meaningful conclusion that SEMA3A inhibits migration/synapse/F-actin inhibition via NRP1. Simply using mutant SEMA3A is far from sufficient.

3) The debating mechanisms for the role of NRP1 in CD8 T Cells (current manuscript versus PMID: 32661362); the latter clearly showed NRP1 is important for “NRP1 cell-intrinsically limited the self-renewal of the CD44+PD1+TCF1+TIM3- progenitor exhausted T cells, which was associated with their reduced ability to induce c-Jun/AP-1 expression on T cell receptor restimulation”. This paper from Vignali group again used the same B16 model to draw a different conclusion but neglected by current manuscript.

4) The switch between cancer types: most mouse models are from B16F10 melanoma that produce SEMA3A and human cancers use renal cancers where SEMA3A is expressed mostly by endothelial cells. Even though switching cancer types may not be a major concern, the histology clearly showed that CD8 T cells are able to extravasate through SEMA3A+ endothelial cell layer (Fig6L). If SEMA3A is able to inhibit F-actin and migration of CD8 T cells, why all CD8 T cell are able to transmigrate through SEMA3A+ endothelial cells? The increased CD8 T cells in SEMA3A region are just simply because there are more blood vessels that carry more CD8 T cells into the tumor microenvironment, irrelevant of SEMA3A status. Again, the claim should be further supported by using the tumors from NRP1 KO mice.

5) Fig2G-I: from comparison to be meaningful, the WT cells should be included. Comparing OE (lentiviral) and KO (PX458 plasmid) are not meaningful due to the different ways of carriers. Lentiviral particle-delivering system is expected to have influence on tumor immunity compared to PX458 (Cas9 may be also problematic). There are just too many variables to make any solid conclusions here.

Minor concerns:

1) Figures 6D, 6F, the last column has no label/legend

2) Line 281: “non-cognate H-2 Kd-gp33 H-2 Kb-Ova pMHC”. Should H-2 Kb-Ova pMHC be deleted?

REVIEWER COMMENTS

We would like to thank the Nature Communication reviewers for the considerable time and effort they have invested into reviewing this manuscript. We have added our comments in blue below each comment.

Reviewer #1 (Remarks to the Author):

In this manuscript, the authors address a contentious issue as to whether SEMA3A is pro-tumorigenic or anti-tumorigenic. They present data suggesting that SEMA3A acts through NRP1 on T cells to increase tumorigenicity, and reduce immunogenicity, by freezing or trapping activated anti tumor T cells in areas of high SEMA-3A expression in the tumor or in the tumor associated vasculature. They correlate extensive murine studies with some patient data showing that in human patients with clear cell renal carcinomas NRP1 T cells were trapped in SEMA3-A rich areas and SEMA3-A expression correlated with reduced survival.

Specific comments:

The authors have provided some elegant studies, principally (but not exclusively) with OT-I T cells that NRP1 is upregulated on T cells when they encounter antigen in an antigen concentration and TCR recognition dependent fashion. Their data shows that NRP-1 deficient mice controlled B16 tumors better than NRP-1 intact mice suggesting that NRP-1 in T cells was causing some sort of inhibition to T cell control. Increased tumor control was associated with increased levels of CD8+ T cells in the tumors. They also knocked out SEMA-3A or overexpressed it in B16ova cells.

The authors then show that the OE and KO B16-ova tumors grow at the same rates in mice. If I am understanding this correctly this seems like a disconnect to me with Fig.2A/B where B16 tumors grow much less well in NRP1 KO mice implying T cell mediated control of these tumors. If Nrp1 KO mice control B16 tumors better than Nrp1 intact mice – dependent upon CD8+ T cells - that implies that B16 growth in these mice is CD8 mediated. In that case would you not expect that the B16-ova tumors with SEMA-3A KO would be controlled much more effectively than OE tumors, especially since the ova tumors have an even better ag to stimulate anti-tumor T cells with? Is this a disconnect between Figs 2A,B and 2G?

Thank you for the positive comments.

The reviewer asks why endogenous NRP1 KO T cells can control tumours better (Figure 2A), then endogenous T cells in mice with SEMA3A knockout tumours (Figure 2G).

We believe this can be explained in two ways:

1) As pointed out by reviewer 2 and by experiments in this paper, there are several biological sources of Sema3A within the tumour. In this paper we show that Sema3A can be produced by hypoxic B16.F10 tumour cells (Extended data Figure 3C) but also by different types of endothelial cells (Extended data Figure 3D), which might affect CD8 T cells ability to infiltrate tumours. For this rebuttal we have explored the exact sources of

SEMA3A in new additional experiments (see below), but the major sources seem to be the B16.F10 tumour cells and endothelial. In experiments concerning Figure 2A/B, the NRP1 floxed/floxed T cells (both CD4 and CD8 T cells) are effectively 'blinded' to all sources of Sema3A; while in the experiments concerning Figure 2G, only one source of Sema3A is removed, namely that from tumour cells. Thus, there will most likely still be other sources of SEMA3A, for example from the vasculature, present in the experiment shown in Figure 2G, which can act on CD8 T cells.

2) Additionally, as we comment in the discussion and as pointed out by reviewer 3, the strong anti-tumour effect we see in Figure 2A/B is likely a combination of NRP1 knockout on both regulatory CD4+ T cells (Tregs) and CD8+ T cells. Others have shown that NRP1 knockout on Tregs *inhibit* Treg function, and that the major ligand for NRP1 on Tregs seems to be Sema4A (see Delgoffe GM, Nature, 2013, PMID: 23913274); while in this manuscript we provide evidence that NRP1 knockout on CD8+ T cells *enhance* their anti-tumor abilities, most likely through Sema3A. These two effects might thus be acting synergistically in Figure 2A/B when NRP1 is removed from both CD4 and CD8 T cells (Tregs are inhibited; CD8 T cells are enhanced); whereas in Figure 2G, only Sema3A is affected.

We thus believe the data is consistent and there are reasonable reasons that endogenous T-cells are better able to control tumour growth in Figure 2A/B than in Figure 2G.

We have added additional data below, section in the discussion about the sources of SEMA3A in reply to a question by reviewer 2 (see below) on line 513-522 of the manuscript which pertains to a similar point and added more background on regulatory T cells, as part of our reply to reviewer 3 (see below).

In this respect much of the pre-clinical data is performed with OT-I T cells. Related to the points above, if B16 tumors are well controlled in NRP-1 KO mice, then it should be possible to show these effects also for truly endogenous anti-tumor T cells. For example, in mice vaccinated against B16 tumors (eg with B16-GM-CSF+anti-CTLA4) endogenous anti-tumor T cell responses will be generated. Under these circumstances how would SEMA-3A KO or OE tumors fare in regular mice? Would the endogenous anti B16 T cell responses control SEMA-3A KO tumors much better than regular B16 tumors or OE B16 tumors? Such data would add confidence that the effects seen with OT-I T cells were also relevant to endogenous anti-tumor T cells recognizing 'real' tumor antigens.

The question, as we understand it, is how SEMA3A KO/OE tumours would grow in animals vaccinated against B16 tumours, in order to explore Sema3A's effect on endogenous anti-tumour T cells.

To avoid misunderstandings, we want to recapitulate that the mice in Figure 2A/B were not treated with adoptive transfer of T cells (OT-I or otherwise) and neither were the mice in Figure 2G – only the mice in Figure 2H were treated with OT-I T cells; however, B16.F10 tumor growth in NRP1 floxed/floxed mice is to a large extent restricted by CD8 T cells as we show in Figure 2C, which uses a similar setup as in Figure 2A. These T cells must thus be endogenous anti-tumor T cells. The reason these T cells can recognize B16.F10 is most likely that the B16.F10 cell-line has some degree of immunogenicity, express a

number of MHC Class I restricted epitopes (Castle et al, Cancer Res, 2012, PMID: 22237626) and can induce an anti-tumour T cell response when not suppressed by other factors, such as PD1 (see for example Shitaoka et al, Can Res Imm, 2018, PMID: 29475880) or possibly regulatory T cells, as pointed out by reviewer 3. Our data shows that NRP1 deletion in T cells is able to bypass some of this suppression.

We are not able to explore this issue any further unfortunately as the laboratory of Professor Vincenzo Cerundolo has closed following his death, and extensive animal experiments are not possible, but we hope that our clarification that key *in vivo* experiments were based on endogenous T cell responses addresses the reviewer's concern.

With KO and OE tumors on different flanks + act of OT-I the KO tumors grew significantly more slowly than the OE tumors with more infiltration of OT-1.

We agree with this comment / conclusion.

Showed that SEMA-3A significantly affects T cell adhesion and mobility and prevents formation of the immunological synapse by confocal.

We agree with this comment / conclusion.

The authors also show that SEMA-3A is associated with poor survival in clear cell renal cell carcinoma patients. Is this association also true in other cancers? Especially those amenable to ICB where presumably anti tumor T cells are available and can be inhibited if SEMA-3A is expressed? One would expect that such an association would be widespread if SEMA-3A expression is critical to T cell control of tumors then there should be a clear association in immunogenic tumors which respond to immune checkpoint blockade and a negative association in those that do not.

This is a very interesting point. We examined all available cancers from TCGA which have enough data on SEMA3A, and compared survival between the top 30% SEMA3A expressing patients with the 30% lowest expressing patients. The results are inserted here and included in Extended Figure 6B.

In Kidney cancer (RCC), cervix cancer and low-grade glioma, and to a lesser extent lung cancer, increased levels of SEMA3A mRNA were associated with a poorer prognosis (right graph). As can be seen by comparing mRNA levels between normal and cancerous tissue, there is considerable heterogeneity in SEMA3A expression between cancer patients (left graph).

Cancer types in which checkpoint inhibitors are used include advanced melanoma (called skin here), lung cancer, kidney cancer, bladder cancer, cervix cancer and Hodgkin lymphoma.

We thus find an overlap with potential immunogenic tumors such as kidney cancer, lung cancer and cervix cancer, but notably, not melanoma.

We note that the main source of SEMA3A in some cancers might be from neovasculature—not necessarily the tumor-cells themselves - in the tumor-microenvironment, as we show in this paper in ccRCC (Figure 6L) and there is thus most likely both an intra-population variability and intra-tumor variability in where the ligand is expressed and what its effects are.

In the main text, we have added the following text:

Line 404-406: “SEMA3A expression was also associated with poor survival in cervix and low grade gliomas (**Extended data Figure 6B**)”.

In patient samples SEMA-3A co-localized with the endothelial marker CD31 and CD8 T cells clustered in the regions of SEMA-3A expression. The authors suggest that this shows that SEMA-3A expression is associated with tumor vasculature and that these areas trap CD8+ T cell in the tumor. If that is true, can they confirm that in their pre-clinical models? Thus, in a B16ova tumor that is seeded with, for example, 20% SEMA-3A OE cells and 80% KO cells one would predict that infiltration of adoptively transferred OT-I T cells would cluster around SEMA-3A OE cells and be reduced elsewhere in these tumors compared to say a 100% KO tumor; similarly the control of a 20% OE/80% KO tumor by OT-I T cells

would then be expected to be worse than a 100% KO tumor with OT-I clustered around the islands of SEMA-3A positivity. These sorts of expts would be very valuable to refute the possible explanation of the human histology that SEMA-3A expression favors T cell infiltration into tumors because that is where the T cells are seen) and that a lack of SEMA-3A expression is actually contributing to the immunological desert -which is counter to the authors' hypothesis.

The reviewer asks us to confirm the findings of SEMA3A localization from RCC immunohistochemistry in our B16.F10 model, and whether CD8+ T cells cluster around SEMA3A rich areas as well.

We performed immunofluorescence (IF) on B16.F10-mCherry positive tumours. Mice were briefly treated with Brefeldin A before harvest of tumours, to inhibit secretion of SEMA3A. Tumours were then stained for CD31, Sema3A and CD8, as well as DAPI (Image F below). We also reanalyzed B16.F10 tumours that were stained for Sema3A, CD31, CD45 and quantified the number of individual cells that were positive for Sema3A using flow cytometry (image E).

Our image analysis indicates that the majority of SEMA3A+ areas are towards the core of the tumour (Image F). Many B16.F10 cancer cells express SEMA3A, as do few CD45+ cells and CD31+ cells. The Sema3A staining is more dispersed in the B16.F10 tumours and not centered specifically around the CD31+ vasculature, as in the ccRCC cancer samples. Our flow analysis shows that in B16.F10 tumours, the majority of SEMA3A is from the B16.F10 cells (image E). However, we note that SEMA3A can be secreted and bind to other cells via glycosaminoglycans on cell surfaces (Wit DJ et al, Mol Cell, Neuroscience, 2005. PMID: 15866045), and so the source and the location of SEMA3A might be different. Concerning CD8+ cells: in area's that are rich in SEMA3A, most T cells are found in / close to SEMA3A. An example image is shown in image F, right side. This is reminiscent of what we see in ccRCC samples.

These data are now included as Supplementary Figure 3E-F.

In response to this comment (and address other comments from other reviewers), line 232-241 now reads:

“While not directly comparable to cells in the lung, flow cytometric analysis of B16.F10 cells grown for 11 days *in vivo* nonetheless showed expression of SEMA3A within the

TME, mainly from tumour cells but also blood endothelial cells (BEC), lymphatic endothelial cells (LEC) and some CD45+ cells (**Extended data Figure 3D-E**). Immunofluorescence staining showed SEMA3A was mainly located around the core of the tumour, with SEMA3A rich areas containing many CD8+ T cells (Extended data Figure 3F). While multiple cell types thus contribute to SEMA3A production within the TME, we generated B16.F10.Ova cells with *Sema3a* knocked out or overexpressed (referred to as *Sema3a* KO and *Sema3a* OE, respectively) as the most straight forward way to experimentally manipulate the levels of SEMA3A levels in the TME.”

The reviewer also suggests mixing SEMA3A OE and KO cell-lines and examine clustering of T cells in relationship to SEMA3A. We agree with the reviewer that this sort of experiment would be highly informative, but also technical extremely challenging: Co-injecting cell-lines this way would probably not generate clearly demarcated areas with only OE or KO cells, but a random mix, and there would be no way to control for the location of Sema3A, as it would be secreted from OE cells and could diffuse into areas with KO cells. As pointed out in response to the reviewers first question – and shown in our analysis of SEMA3A sources above – there are also several other sources of Sema3A within the TME, including endothelial cells, which play a major role in determining the level of infiltration into tissues. As we show in this study (Extended data 3D-F), a large proportion of these cells express SEMA3A within the TME as well.

The manuscript currently contains from multiple orthogonal experiments (Figure 3 and 5) showing that Sema3A can inhibit NRP1+ CD8+ T cells migration. While these experiments are done *in vitro*, they are well-controlled, and allow us to specifically examine SEMA3A's effect on T cell migration. In combination with new data presented in the rebuttal, we believe it is likely that SEMA3A plays similar roles in both B16.F10 and ccRCC tumours. Our explanation is also in line with Leclerc et al which also show that SEMA3A inhibit T cell migration (Figure 2D-E, Leclerc et al, Nature Comm, 2019, PMID: 31350404) and resembles what is seen with macrophages trapped in hypoxic tissue through the SEMA3A/NRP1 axis (Casazza et al, Cancer Cell, 2013, PMID: 24332039).

We have added the following sentence to the discussion:

Line 480-486: “These experiments are in line with findings from Leclerc et al, which also find that SEMA3A can inhibit T cell chemotaxis toward CXCL12 in transwell assays (19).”.

Have the authors tried to use their data for therapy –showing that NRP-1 is an important checkpoint by using an antagonistic NRP-1 antibody *in vivo* – that is different from using the KO and OE cells used here which cannot provide a therapeutic outlet for this work in the clinic.

We have not tried to use antagonistic NRP1 antibodies in a therapeutic setting in our tumour models, but did use anti-NRP1 antibodies for the experiments outlined in Figure 3D (on T cell migration). A number of high-quality articles have shown that antagonistic NRP1 antibodies enhance anti-tumour immunity *in vivo*, probably through multiple mechanisms: for example Delgoffe et al (Delgoffe et al, Nature, 2013, PMID: 23913274) have shown that NRP1 antibodies can affect NRP1-expressing Tregs and tumour growth of B16.F10. Leclerc et al (Leclerc et al, Nature Comm, 2019, PMID: 31350404) examined NRP1 and

PD1 antibodies in combination and showed that they could decrease B16.F10 tumour growth and increase CD8 T cell infiltration. NRP1 also plays a role on macrophages (Casazza et al, Cancer Cell, 2013, PMID: 24332039), is expressed on neovasculature and affects tumour angiogenesis, which can be affected by anti-NRP1 antibodies (Pan et al, Cancer Cell, 2007, PMID: 17222790), and is even found directly on certain cancer cells (Rizzolio et al, JCI, 2018, PMID: 29953416). Anti-NRP1 antibodies will likely affect all these other important determinants of tumour growth as well, and so would not directly answer the question of the Sema3A/NRP1 axis on T cells, which is the main story of this manuscript. We therefore felt using T-cell-specific (CD4-Cre) knockout of NRP1 was more likely to directly answer questions on what role the receptor has on CD8 T cells.

In our discussion we have added the following text:

Line 549-551: “These results are in line with Delgoffe et al. (63) and Leclerc et al. (19) who show similar control of tumour growth when treating mice with a blocking anti-NRP1 antibody.”.

In summary, this is an extensive manuscript with well controlled and well performed experiments. The data show that SEMA-3A may play an important role in limiting T cell mediate tumor immunity and the experiments shown are largely convincing. In my opinion, the conclusions could be significantly strengthened by generalizing the results away from just the use of the OT-I system; by extending the importance of the correlative data from ccRCC to other human tumors, by showing that what is observed in human tumors holds true in the pre-clinical models in terms of the association of T cells with SEMA-3Ahi regions of tumors to show that SEMA-3A is truly inhibitory rather than stimulatory for T cell infiltration and by showing how these data could be used clinically in a pre-clinical model (ie with an anti-NRP-1 Ab type strategy). That said, there is already a large amount of data included and the authors may be able to address some of these concerns without necessarily demanding extensive new data.

Thank you for these comments.

In terms of expanding the analysis: Besides using OT-I T cells, we do also show similar effects using 3LL tumour models (Figure 2B) and ccRCC (Figure 6). As indicated in the reply to comments above, we've now also expanded the analysis to look at other TCGA datasets (Extended Figure 6).

Reviewer #2 (Remarks to the Author):

The manuscript written by Dr. Mike B Barnkob, et al. reveals novel function of SEMA3A in TME. SEMA3A acts as an inhibitor of tumor associated CD8+ T cells via its receptor Nrp1, leading to cellular paralysis. Their theory was also confirmed by examining a clear cell renal cell carcinoma patient, found that SEMA3A expression is associated with reduced survival and CD8+T cells accumulation. The manuscript is well structured and substantial experiments are conducted. Most of them seem reasonable and I basically feel positive for publication of this work. However, some important points should be addressed and

clarified. Also, there are many careless mistakes in figures. I recommend authors should review the details of manuscript carefully and revise it to correct any tiny mistakes (As far as I noticed, I described in the section <minor remarks>).

Thank you for these positive comments.

We apologize for the mistakes in the figures, which seem to have arisen during conversion from letter format to A4. These have been corrected.

<Major remarks>

1) In Fig. 1G and Sup Fig. 1D, authors identify the main receptor as PlexinA1 because of its high expression on CD8+T cells. However, as described in Line 120, PlexinA4 is also expressed on CD8+T cells. Previous reports have shown that SEMA3A-PlexinA4, unlike PlexinA1, activates macrophage inflammation (JEM 2010). As such, PlexinA4 may send signals in a different direction than PlexinA1. If authors think PlexinA4 is not involved in the theories of this paper, please prove it (e.g. by using knockdown or knockout cell lines), or at least it should be discussed.

We agree with the reviewer that Plexin-A4 might play a role as well as Plexin-A1 and have corrected the text as indicated below. Recent publications also show that Plexin-A4 plays an important role on T cells, and we have edited the text to reflect this (Celus W et al, Cancer Imm Research, 2022, PMID: 34815265).

It is clear from the literature that NRP1 can partner with both receptors, probably with the same affinity. In the paper we focus on Plexin-A1 as this has such a clear upregulation following activation on T-cells, but the reviewer is correct that the effects we see from Semaphorin 3A on T cells could be explained through both NRP1:Plexin-A1 and NRP1:Plexin-A4 receptor complexes. Plexin activation and signaling require both binding of semaphorin to its extracellular domain (an interaction that is greatly enhanced by NRP1, see Janssen et al, Nat Struct Mol Biol, 2012), and binding of RhoGTPase to its intracellular RhoGTPase-binding domain (RBD) (Wang et al, Science Signal, 2012, PMID: 22253263). It's worth noting that Plexin-A1 and A4 are very similar both in terms of their intra-cellular signaling domains and both form an almost identical ring-like structure of their extracellular domains (Kong Y et al, Neuron, 2016). Alignment of their cytoplasmic domains (aa 1264-1894 for Plexin-A1 & aa 1258-1893 for Plexin-A4), reveals 82% sequence homology. Consistent with this, both proteins have been shown to have identical RapGAP activity (Wang et al, Science Signal, 2012, PMID: 22253263).

Because of the above discussion, we have made the following edits in the text:

Line 49-51: "Here we demonstrate that Neuropilin-1 (NRP1) and Plexin-A1 and -A4 are upregulated on stimulated CD8+ T cells, ..."

Line 103-104: "In this study, we report that NRP1 and Plexin-A1 and -A4 are upregulated following activation of CD8+ T cells corresponding to the level of TCR stimulation."

Line 142-147: "Having identified NRP1, Plexin-A1 and Plexin-A4 receptors on stimulated CD8+ T cells, we determined if these cells bind SEMA3A (16)."

Line 383-384: "SEMA3A signaling through Plexin-A1 and -A4 inactivates the small GTPase Rap1A (47), which in turn can modulate myosin-IIA activity in diverse cell types (48, 49)"

Line 589-592: the following text have been added: "While we mainly focus on Plexin-A1 in this study, it should be noted that the effects we see from Sema3A on T cells could be explained through both NRP1:Plexin-A1 and NRP1:Plexin-A4 receptor complexes. Plexin-A1 and -A4 are highly similar both in terms of their intra-cellular signaling domains and both form an almost identical ring-like structure of their extracellular domains (68). Consistent with this, both proteins have been shown to have identical RapGAP activity (69)."

2) In Fig. 2 and Sup Fig. 3, results between flu infection model and tumor model are different. Authors explained that tumors and endothelial cells strongly express SEMA3A in hypoxic TME. Regarding tumor cells, it is reasonable because SEMA3A expression level is up-regulated under hypoxia and SEMA3A-KO tumors grow slower. Regarding endothelial cells, percentage of SEMA3A positive cells in tumor LEC/BEC (Sup Fig.2E) is larger than that of infected lung (Sup Fig.2B). However, whole lung LECs and BECs in pneumonia and LECs and BECs in the tumor environment may be essentially different.

We agree with the reviewer that it might be hard to directly compare between the lung and an injected tumour cell-line. We have clarified the text as follows:

Line 232-235: "While not directly comparable to cells in the lung, flow cytometric analysis of B16.F10 cells grown for 11 days *in vivo* nonetheless showed expression of SEMA3A within the TME, mainly from tumour cells but also blood endothelial cells (BEC), lymphatic endothelial cells (LEC) and some CD45+ cells. (**Extended data Figure 3D-E**)."

2-1) Do authors simply see the difference of SEMA3A expression between mature endothelial cells that are originally present in lung and neoplastic endothelial cells? Or do authors think SEMA3A expression in LECs/BECs is also induced by hypoxia? Please describe in the manuscript.

Most likely, the difference we see in SEMA3A expression between endothelial cells (ECs) in tumors (Supplementary Figure 2E) and lung (Supplementary 2B) is due to the latter consisting of mature EC's, and the former of developing vasculature.

Klomp et al (Klomp J et al, J Biol Chem. 2020), have elegantly studied the effects of hypoxia on an EC model system (cultured HUVECs) at different oxygen tensions (from 0.1, 1, 3, 5, up to 21%), between 2-48 hours after induction of hypoxia. We re-analyzed this data to look for SEMA3A expression. In this model, SEMA3A transcript level is only weakly affected (at between 18-24 hours), with a slight down-regulation. Similarly, there is some evidence in the literature that SEMA3A is down-regulated in coronary arteries in hypoxic conditions (Narematsu M et al, J Mol Cell Cardiol, 2020) but this area in general seems under-explored.

Conversely, it is well established that SEMA3A is expressed by endothelial cells during angiogenesis when new vasculature is developed from pre-existing ones (Serini G et al, Nature, 2003) and play an important autocrine role in angiogenesis and vascular formation by regulating endothelial integrins (Serini G et al, Nature, 2003 & Ochsenbein AM et al, Development, 2016). Much of the vasculature and lymphatics present in B16.F10 tumours are newly developed as these tumours are injected intradermally, and very little vasculature exists initially. As is visible in Supplementary Figure 2G, top figure, this changes quickly in B16.F10 tumours: by day 15 existing blood vessels are massively expanded and multiple new ones are found in the tumour itself.

There therefore seems to be most evidence for the first hypothesis in our model, namely that SEMA3A is differentially expressed between mature endothelial cells in the lung (which express little) and newly developed endothelial cells in the tumours (which express a lot), as is clear from Supplementary Figure 3E. SEMA3A have also been shown to help normalize the vasculature in tumours (Maione F et al, JCI, 2012) and so the SEMA3A we find here might be an insufficient attempt to correct this; or simply due to developmental programs being re-activated in the tumour endothelial cells.

To recapitulate: other studies do not find hypoxia induces SEMA3A in mature EC's, whereas there is evidence that SEMA3A is expressed during angiogenic events and in newly developed ECs. Most likely the SEMA3A we see in tumor ECs is because these are newly developed ECs and/or because of re-organization of the tumor vasculature.

We have added the following to the discussion:

Line 516-525: "The source of SEMA3A in the TME are likely multiple. We show here that SEMA3A can be expressed by BEC and LEC cells in tumours, by some immune cells (**Extended data Figure 3D-E**) and also by the tumour cells themselves conditions (**Extended data Figure 3C and 3E-F**). Conversely, little to no expression was seen in draining and non-draining lymph nodes. Some immune cells, especially CD4+ T cells and dendritic cells, have previously been shown to upregulate SEMA3A when activated (61). SEMA3A is also known to be expressed by endothelial cells during angiogenesis (13) where it plays an important autocrine role in vascular formation by regulating endothelial integrins (62). SEMA3A might thus be upregulated as neo-angiogenesis happens in the tumor or as an attempt to normalize the vasculature (29)."

2-2)

-if yes for former question in 2-1)-

In pneumonia, there are some localized hypoxic environments at inflammatory foci. Show IHC or other convinced data looking at local inflammatory foci to explain even in hypoxic-estimated environment, SEMA3A expression is still low and CD8+T accumulation is normal in the lung.

We are interpreting the question referred to here as the following, from the reviewers question in 2-1: "Do authors simply see the difference of SEMA3A expression between mature endothelial cells that are originally present in lung and neoplastic endothelial cells?".

As mentioned, and referenced above, we believe the difference is mainly related to how recently ECs in the lung (mature) and tumor (new) were developed. Newly developed and developing ECs are known to express SEMA3A. This also means we do not necessarily expect mature ECs to upregulate SEMA3A in local hypoxic environments in the lung. We do not understand the question about hypoxic loci in the lung.

More generally, while we appreciate the reviewers' points concerning the difference between BECs and LECs in hypoxic and non-hypoxic regions in the lung, we are cautious not to make strong conclusions along this line either way. First, it is not technical possible for us to induce and examine localized hypoxic regions in the lungs and then explore CD8 T cell migration into these. For example, we are not sure the influenza strain we use (which is specifically chosen because it does not lead to an antibody response) also induce localized hypoxic regions in the lung. Secondly, we are not making any comments about hypoxia in ECs in the paper. While a discussion concerning upregulation (or not) of SEMA3A in mature endothelial cells is interesting, for this manuscript we think the above text added (under question 2-1) to the discussion is what is reasonable supported with data presented here.

- if yes for latter question in 2-1)-

Show the data that the elevation of SEMA3A in endothelial cells is induced by hypoxia and discuss why it does not happen inflammatory hypoxia foci in pneumonia.

Please see answer above, which we hope satisfies this point.

2-3) what is the molecular signal mechanism under hypoxia condition that can induce SEMA3A expression in tumor and/or other cells?

This is an interesting question. Here we find that SEMA3A can be upregulated in tumor cells under hypoxic conditions (Extended Figure 3C). Others have also found SEMA3A expressed in many cancer cell-lines (Catalano, Blood, 2006), and found in PIMO-positive (hypoxic) cancer cells specifically (Casazza A et al, Cancer Cell, 2013). In the newly added Extended Figure 3F, SEMA3A does also seem to be primarily expressed near the necrotic core of the tumour. It is tempting to think this region is also more hypoxic. Most likely it involves HIF1a, which acts as a master regulator for hundreds of genes under hypoxic conditions. We identify key genes associated with HIF1a as upregulated in B16.F10 cells when these are based under hypoxic conditions (*Bnip3*, *Vegfa*, *Pkd1*), together with SEMA3A (Extended Figure 3C). Again, while a discussion concerning upregulation (or not) of SEMA3A in different cells is very interesting, we believe it is beyond the scope to address here.

These questions need to be addressed to clarify whether the theory in this paper is truly tumor-specific or not.

3) SEMA3A is produced by not only tumors and endothelial cells. For example, CD16+ monocyte is one popular source of SEMA3A in inflammatory condition. I understand Fig.2 C-E clearly show the effect is mainly involved in CD8+T cells, but authors should show the profiles of SEMA3A expression in other key players in TME (either or both human/mice

TME) to increase the generality of this work.

Extended data figure 3E now shows SEMA3A expression on CD45+ cells as well, with around 10% of these cells positive for SEMA3A. Line 232-238 has been rewritten in response to this reviewer and questions from reviewer 1 to more clearly state which cell-types express SEMA3A. As noted above, the major source within B16.F10 tumours seem to be the tumour cells themselves.

Additionally, as mentioned above in reply to question 2-1, we have written a new paragraph to more precisely discuss the different sources of SEMA3A reported in this manuscript in relations to other publications.

4) In Fig6 K-M, how do authors separate the SEMA3A Rich from the Poor? Authors describe they construct mathematical scripts; This would be not a popular method. I'd like to request clear and brief explanation how to do it, with visual examples. I mean, I want to know whether this method is sufficiently satisfactory for general readers.

Concerning how SEMA3A rich areas were separated from poor areas:

A trained histopathologist (P.M) reviewed the images and selected the area with the most and least CD31 staining, in a blinded fashion to the SEMA3A and CD8 slides. The extent of staining was quantified in the regions where expression was deemed to be highest and lowest (% stained area = positively stained pixels/total pixels x 100). This analysis was repeated in the same regions on adjacent serial sections for SEMA3A (% stained area = positively stained pixels/total pixels x 100) and CD8 (discrete cell counts were calculated from positively stained regions using a watershedding process). Areas were then cropped and aligned the corresponding areas on the SEMA3A and CD8 slides and we quantified the staining of each marker and correlated the results.

The method was recently described in Bull J et al, Sci Rep, 2020 (PMID: 33122646), and shows accurate cell counting when compared to a number of other methods, including human quantification, ImageJ and Visiopharm (see supp. Figure B1 in cited paper) and are now available online, as written below. We thus feel like the methods are explained in quite an extensive manner now. Images and visual examples can be found in Bull et al (PMID: 33122646), and Vipond et al, PNAS, 2021 (PMID: 34625491). We have added the following text to the method section:

Line 920-929: "Image analysis were performed by a trained histopathologist (P.S.M.) who reviewed the images and selected the area with most and least CD31 staining in a blinded fashion to the SEMA3A and CD8 slides. The extent of staining was quantified in the regions where expression was deemed to be highest and lowest (% stained area = positively stained pixels/total pixels x 100). This analysis was repeated in the same regions on adjacent serial sections for SEMA3A (% stained area = positively stained pixels/total pixels x 100) and CD8 (discrete cell counts were calculated from positively stained regions using a watershedding process). Analysis scripts used here have been described in Bull JA et al (86) and are available at <https://github.com/JABull1066/ImageAnalysisScripts/tree/main>."

<Minor Remarks>

Fig Sup 1a: Briefly describe the method how to do the data mining from public database.

This is described in the methods section under “Analysis of publicly available transcriptional data” (paragraph starting line 662).

Fig 1b: Vertical axis numbers are missing.

Fixed.

Fig 1d: Which is naïve, which is stimulated?

Bars in red are stimulated, and the bar in blue is naïve. Additional legend has been added to make this more clear.

Fig 2c: Blue and red graph should mean Flox/Flox. “ns” is missing.

Yes, fixed.

Fig 2d: Blue dot should mean Flox/Flox.

Yes, fixed.

Fig 2G: Horizontal axis numbers are missing.

Fixed.

Line 183: not 4M, it should be 2M.

Fixed.

Fig 3H: Vertical axis numbers are missing. Also, some reagents in horizontal bar are missing. CXCL10, SEMA3A, what and what?

Fixed.

Fig 4B: Vertical axis numbers are partially missing.

Fixed.

Fig 6FIJ: Vertical axis numbers are partially missing.

Fixed.

Fig 6DF: clarify blue dots mean tumor adjacent tissue.

Fixed.

Reviewer #3 (Remarks to the Author):

The current article has attempted to establish a SEMA3A-NRP1 immune checkpoint on CD8 T cells by interrupting F-Actin and CD8 T cell migration in the tumor microenvironment. There is no doubt that the find could be potentially important due to the novel link from cancer cell-produced SEMA3A to suppress NRP1+ CD8 T cells by inactivating F-Actin and immobilization of CD8 T cells. The article at its current format, however, has some major flaws.

We wish to thank the reviewer for the comments on potential importance of these findings and for the engaging critique below. The comments have made us re-think and re-write proportions of the text and, we believe, enhance the manuscript.

As the discussion is mainly about NRP1s role on Treg and how our work can be understood in the context of the existing literature on NRP1 and T cells, we have made a visualization of our current understanding. It's inserted below to better allow the reviewers to follow and critique our thinking.

Briefly, our understanding is the following: we believe SEMA3A affects CD8+ T cells via NRP1 – as laid out in this manuscript – but that these effects are partially masked when Treg's are still present; but can be “released” when Tregs are affected (via NRP1 ablation, as shown by Delgoffe GM et al., Nature, 2013, PMID: 23913274) or when additional boosts are provided to CD8+ T cells (e.g. via PD-1 blockage, as shown by Chang Liu et al, Nature Imm, PMID: 32661362).

[FIGURE REDACTED]

Model. A comparison between different model systems used to knock-out NRP1 on T cells and indication about which cell and at what stage the primary effect of NRP1-deficiency will be. Note that in our model (CD4-Cre), the tumor inhibition we are documenting, is likely an affect via both inhibiting regulatory T cells (Tregs) and promoting CD8+ effector cells.

Major critiques:

1) No explanation of discrepancy with some very critical publications in the NRP1 field related to Tregs and CD8 T cells, which leads to the potentially wrong interpretation of tumor immune phenotype from CD4-cre/NRP1fl/fl mice published by the Helfrich group (PMID: 23045606). The published JEM paper used the same model and showed that Nrp1+ Tregs are the most critical cell types for tumor promotion, leading to the suppression of CD8 T cells

Thank for highlighting the Hansen et al (JEM) paper. We do indeed use the same model and the reviewer is correct in comparing our finding with theirs. We comment on their paper below, but want to highlight up front that we are not trying to claim NRP1 does not play a role on Tregs or that Tregs are not important.

Whereas Hansen et al (JEM) looks at CD4 regulatory T cells and NRP1's role on Tregs, we focus on CD8+ T cells, as these are less explored (a point highlighted by the other reviewers and as noted in some of the papers mentioned by reviewer 3). We show that

NRP1 is upregulated on stimulated tumor-specific T cells (using multiple different methods in both human and mice). Hansen et al are very specific about why they do not consider CD8+ T cells in their paper, where they write: “Ablation of Nrp-1 expression in Nrp-1KO mice was specifically restricted to CD4+ T cells, as only a minor population of CD8+ T cells (1%) from Nrp-1WT mice expressed Nrp-1”. They come to this conclusion, because they only examine naïve CD8+ T cells, which, as we show here (Figure 1) do not express NRP1. However, today there can be little doubt that activated CD8+ T cells do indeed express NRP1 following stimulation as we and multiple other groups have shown (Leclerc, Nat Comms, 2019, PMID: 31350404 ; Liu C, Nat Imm, 2020, PMID: 32661362; JB Williams et al, JEM, 2017, PMID: 28115575 ; M Philip et al, Nature, 2017, PMID: 28514453). Hansen et al also concludes that “This entire process [NRP1 ablation in T cells] resulted in a compensatory enhanced activation of recruited CD8+ T cells within these tumors...”. This intriguing finding could possibly be explained by some of our results here, namely that NRP1 deletion on CD8s also free them from additional suppressive mechanisms through SEMA3A. NRP1 likely plays different roles on regulatory and effector T cells (as we expand on below), and we thus feel close and careful examination on both cell types are warranted.

For this reason, we generally feel the two stories are complementary, examining different, but equally important, aspects of T-cell biology in different cell-types. It is perhaps useful to think about the timing and where these cells play a major role: Tregs would be expected to be particularly important doing priming of the anti-tumor response (by inhibiting DCs, suppressing effector cells in the lymph node, and producing inhibitor cytokines), whereas CD8's are acting down-stream of this process, and are the effector arm of this response. Both aspects are important and both cells affect tumour progression at different locations and time-points. For example, when we remove CD8 T cells in NRP1-floxed mice, using depleting antibodies, the anti-tumor effect disappear (Figure 2C); In Liu et al (PMID: 32661362), where only CD8+ T cells have NRP1-deletions, tumour growth is further inhibited when mice are treated with anti-PD1 therapy, as discussed below – clearly CD8 T cells and NRP1+CD8+ T cells affect tumor growth when using the B16F10 and 3LL models.

In our discussion we originally wrote: “Most likely, the remarkable control of tumour growth seen by us and others is due to synergistic effects [between Tregs and CD8s], as our data would suggest that ablation of Nrp1 enhances CD8+ T cell migration and effector functions as well.” We are thus not trying to discount Tregs, and we do not see much discrepancy between our interpretation here and the articles referenced by the reviewer.

From these comments it is clear, that the current text is not clearly enough formulated, and we have updated the text as indicated in the reply to the questions below.

Specific deletion of NRP1 in CD8 T cells (E8I cre, only deplete NRP1 in CD8 cells, PMID: 32661362) has no phenotype in primary tumor growth (B16 melanoma, the same model used in this paper). This further supports the primary tumor phenotype in CD4-cre/NRP1fl/fl mice is due to Tregs. Further clarification is required since this will directly impact the significance of this study: whether or not SEMA3A has its impact on CD8 T cell via NRP1.

Concerning *Sema3A*'s effect on CD8+ T cells, we will address this point below, in reply to question 2 by the reviewer below.

Concerning Liu et al paper (which uses a E81 Cre-NRP1 model): We do not believe Liu et al claim that their tumour phenotype is primarily due to Tregs. The paper describes a phenotype specific in memory CD8+ T cells, showing that NRP1 T cells are more exhausted – in line with what we hypothesize concerning the CD8+NRP1+PD1+ TILs we identify in ccRCC patients – and that NRP1 on CD8 T cells can negatively affects their memory response and thus ability to fight cancer cells and proliferate in a recall model. Many of the findings in their paper match ours, including the fact that NRP1 is upregulated on tumor-specific T cells together with PD1. Importantly, the authors find that NRP1 deletion on CD8's did have an effect on the growth of primary tumours if these were also treated with PD1 antibodies, to a higher degree than PD-1 treatment alone did (see Figure 1I). This shows that NRP1 does indeed play a modulatory role on CD8's during primary anti-tumor response.

However, the reviewer is correct that tumour growth in their E81-Cre-NRP1-floxed mouse model is distinct from what we see in our CD4-Cre-NRP1-floxed mouse model, since we see an effect on tumor growth without anti-PD1-treatment. While we remove NRP1 on both CD4 and CD8 T cells, Liu et al use a CD8-specific Cre-promotor, but otherwise use the same tumor cell-line, inject a similar number of cancer cells in the same anatomical region we do in this paper. Using a CD8-specific promotor might also affect CD8+ dendritic cells, so there could be subtle but important difference between our models. But it does seem like a discrepancy that tumor is severely inhibited in our (and Hansen et al's) model, and not theirs following NRP1 knockout.

Still, we believe a way to reconcile the difference in tumor growth between models is by considering especially the priming phase of CD8 T cells, as outlined in the figure above. This phase happens in lymph nodes and might still be stifled in E81-Cre-NRP1 mice, where NRP1+ Tregs might be able to suppress the priming of tumor-specific T cells, since NRP1 is still expressed on these cells. NRP1's role in enhancing Treg's and their ability to suppress CD8s has been documented by the same group (Delgoffe GM et al., *Nature*, 2013, PMID: 23913274; Overacre-Delgoffe, A. E. et al, *Cell*, 2017, PMID: 28552348). It is worth considering that the B16 tumor model develops quite rapidly (over 14-20 days), making it hard for the adaptive immune system of the mouse to mount a sufficient response quickly enough: most likely, any tumor-specific T cells generated during this initial period of priming and expansion (which takes about 7-10 days) is too impotent to seriously impede tumor growth without help in the form of anti-PD1 (as Liu et al show), or in our case, by less Tregs-inhibition via NRP1 ablation. Conversely, *SEMA3A* over-expression can also modulate this response in the opposite direction, as we show in Figure 2I.

To recapitulate, we believe the differences between models might be due to NRP1's different effects across multiple cell-types (regulatory T cells and effector CD8 T cells) and in different context (priming in lymph node, effector functions in tumors and subsequent memory formation).

We have added the following text throughout the paper to make clear that we are deleting NRP1 on both CD4 and CD8 T cells, and to further discuss our findings in relationship to the published literature on NRP1 on Tregs:

Line 52-54: "Deletion of NRP1 in both CD4+ and CD8+ T cells enhanced CD8+ T-cell infiltration into tumours and restricted tumour growth in animal models."

Line 88-90: "SEMA3A binding to Plexin-A requires the co-receptor NRP1 (15, 16), which is found on CD4+ regulatory T cells (Tregs) (17, 18) and tumour-infiltrating CD8+ T cells (19, 20).

Line 168-172: "Prior studies in the B16.F10 tumour model demonstrated that anti-NRP1 antibodies enhance CD8+ T cell infiltration and reduce tumour growth, but that knockout of NRP1 only in CD8 T cells had an effect only when combined with anti-PD-1 antibodies (26). This suggested that NRP1 function on CD4 T cells might mask the effects NRP1 loss on CD8+ T cells."

Line 551-581: "Hansen et al also found strong anti-tumour effects in a comparable conditional *Nrp1* knockout model and ascribed this primarily to decreased Treg infiltration into the TME (65). However, using a *E8/Cre*-NRP1 model, to specifically remove NRP1 on mature CD8+ T cells, Liu et al, only found an effect on tumor growth, if an anti-PD1 antibody was used co-committedly (26). Why do we and Hansen et al (65) see a strong effect on primary tumour growth, while Liu et al (26) did not? Most likely, the remarkable control of tumour growth seen by us and others is due to synergistic effects between NRP1 knockout on both regulatory T cells and CD8+ T cells. As shown in this manuscript, ablation of *Nrp1* enhances CD8+ T cell migration and effector functions by inhibiting SEMA3A binding. Conversely, research by Vignali and colleagues has shown that NRP1 plays a key role in Treg survival and suppressive capabilities through ligation with Sem4A (63, 66), and so affecting NRP1 on both T cell subsets likely allows the immune system to better control tumor growth, than removal of NRP1 on CD8+ T cells alone can. We thus speculate that the effects of SEMA3A on CD8+ T cells are predominant when these effector cells are not inhibited by NRP1+ Tregs, as indicated here and by Hansen et al (65), or freed from co-inhibition using anti-PD1 therapy, as shown by Liu et al (26). Indeed, when we only affect SEMA3A expression in tumour cells (via overexpression or deletion), tumour growth was not affected, consistent with the hypothesis that attenuation of Treg's is needed to fully release CD8+ effector T cells."

2) The SEMA3A and NRP1 connection on CD8 T cells is not fully established. In addition to the concerns raised in 1), the functional connection between SEMA3A and NRP1 within CD8 T cells are not fully established. Figs3-5 aim to study the function of SEMA3A in CD8 T cells. NRP1 KO CD8 T cells should be used and compared to draw a meaningful conclusion that SEMA3A inhibits migration/synapse/F-actin inhibition via NRP1. Simply using mutant SEMA3A is far from sufficient.

We respectfully disagree that SEMA3As effects on CD8 T cells have not been established. To recapitulate our data: we show that the SEMA3A receptors (NRP1 and Plexin-A's) are present on tumour-specific CD8+ T cells (Figure 1); that SEMA3A is present in the TME (Extended Figure 3C-E); that SEMA3A can bind stimulated CD8s (Figure 1E & Figure 4C);

that SEMA3A can affect the cytoskeleton of CD8 T cells, and their ability to form synapses and migrate, two major T cell phenotypes (Figure 3-5); and that SEMA3A overexpression or KO, leads to changes in CD8 T cell migration (Figure 2F and 2I). Additionally, we show that the anti-tumour effects are removed when CD8+ T cells are removed from NRP1 KO mice with depleting antibodies (Figure 2C); and that NRP1-deficient CD8+ T cells migrate better into tumour when using bone-marrow chimeras (Figure 2F).

Our data is also in agreement with other papers which also examine SEMA3A signaling via NRP1:Plexin-A's including articles concerning SEMA3A's effect on CD8+ T cells. There is a rich literature from multiple cell-types showing that Semaphorin bind Plexin-A's (Winberg ML et al, Cell, 1998, PMID: 9875845 & Takahashi T et al, Cell, 1999, PMID: 10520994), and that NRP1 enhances this binding (Janssen et al, Nat Struct Mol Biol, 2012). The biological signaling pathway is well elucidated (Wang et al, Science Signal, 2012, PMID: 22253263). In the context of anti-tumor immunity, our work is also in agreement with other studies indicating a role of NRP1 on CD8+ T cells and that SEMA3A can indeed affect CD8+ T cells negatively (Leclerc et al, Nat Comms, 2019, PMID: 31350404; Lepelletier Y et al, Eur J Imm, 2006, PMID: 16791896; Catalano A et al, Blood, 2006, PMID: 16380453).

If we understand the reviewer correctly, we are asked to redo the experiments in figure 3-5 using NRP1 KO CD8+ T cells. To do as the reviewer asks, we would have to cross CD4-Cre, NRP1-flox miced and OT-I mice, to generate NRP1 KO OT-I specific T cells. We completely agree with the reviewer that such a model would provide a third way to examine the effects we see (besides using blocking antibodies and mutant SEMA3A recombinant proteins). We did spend considerable time and resources attempting to make this cross, but for technical reasons this mouse never fully succeeded. This was due to changes in OT-I expression between the models, which would make comparisons imprecise. The OT-I TCR was originally inserted randomly into C57BL/6 blastocysts, meaning that crossing multiple strains together can lead to dilution of the TCR, and in our cases our mice ended up with substantially less expression of OT-I TCRs. This year-long request is not possible for us to undertake again.

We do use both blocking antibodies against NRP1 (as has also been used by the other papers commented on by the reviewer above) and recombinant SEMA3A (mutant SEMA3A) which specifically lack the moieties that enable binding of the protein to NRP1 in many of the experiment, because we do believe these reagents are sufficient to make the conclusions that we do. These reagents have been used in a number of articles published in high-impact journal before (Mutant SEMA3A: Janssen BJC et al, Nat Struct Mol Biol, 2012; Blocking NRP1 antibodies: Delgoffe GM et al, Nature, 2013). Others in the field use similar reagents, and we therefore respectfully disagree that they are insufficient.

Taken together we therefore do feel like there is compelling evidence that SEMA3A indeed can affect CD8+ T cells.

3) The debating mechanisms for the role of NRP1 in CD8 T Cells (current manuscript versus PMID: 32661362); the latter clearly showed NRP1 is important for "NRP1 cell-intrinsically limited the self-renewal of the CD44+PD1+TCF1+TIM3- progenitor exhausted

T cells, which was associated with their reduced ability to induce c-Jun/AP-1 expression on T cell receptor restimulation". This paper from Vignali group again used the same B16 model to draw a different conclusion but neglected by current manuscript.

We do reference this paper (Liu et al, which we discuss above; reference 23 in the old manuscript), and are not trying to neglect this paper. As written above we have added further results from their paper to our discussion in order to further this discussion. However, since Liu et al's paper is focused on memory CD8 T cell response (something we do not explore), whereas ours is focused on other aspects of T cell biology. Again, Liu et al do find an effect on the primary anti-tumor response when anti-PD1 antibodies are used. Since we find strong effects of SEMA3A on IS formation, this might explain why NRP1 KO cells are better able to mount a recall response in Liu et al's paper. It is worth noting that NRP1 has most likely not evolved as a marker of exhausted T cells, but due to its ability to bind other ligands and receptors. We are more focused on this latter ability in this manuscript, but how NRP1-mediated signaling can perhaps lead to the phenotype on memory T cells is very interesting, but outside the scope of this manuscript.

4) The switch between cancer types: most mouse models are from B16F10 melanoma that produce SEMA3A and human cancers use renal cancers where SEMA3A is expressed mostly by endothelial cells. Even though switching cancer types may not be a major concern, the histology clearly showed that CD8 T cells are able to extravasate through SEMA3A+ endothelial cell layer (Fig6L). If SEMA3A is able to inhibit F-actin and migration of CD8 T cells, why all CD8 T cell are able to transmigrate through SEMA3A+ endothelial cells? The increased CD8 T cells in SEMA3A region are just simply because there are more blood vessels that carry more CD8 T cells into the tumor microenvironment, irrelevant of SEMA3A status. Again, the claim should be further supported by using the tumors from NRP1 KO mice.

The reviewer asks why some CD8+ T cells in the ccRCC tumours are able to transmigrate.

As we show in Figure 6D, not all CD8+ T cells in ccRCC tumours are NRP1 positive (with an average around 20-25% of all CD8+ TILs being NRP1+). As per the rest of the paper, we believe NRP1 plays a role in sensitizing CD8+ T cells to SEMA3A, so without this receptor present, CD8+ T cells are not affected in the same way and are able to bypass the effects of SEMA3A in ccRCC. In situations where all T cells express NRP1, the effect is more stark, as we show in Figure 3D-E and 3H. Here SEMA3A seriously affects T cells ability to migrate, an effect that can be inhibited by using blocking NRP1 antibodies (Figure 3H). Some of these experimental designs are similarly to the effect Leclerc et al use, and use the same blocking NRP1 antibody that the Vignali group uses in Delgoffe, G. M. et al., Nature, 2013.

Concerning experimental proof from mice, please see IF image concerning localization of T cells in B16.F10 tumors in response to reviewer 1 above. We are unfortunately not in a position to repeat these experiments with NRP1 KO mice, due to the closure of the laboratory in which these strains were kept, following the death of the senior author.

5) Fig2G-I: from comparison to be meaningful, the WT cells should be included. Comparing OE (lentiviral) and KO (PX458 plasmid) are not meaningful due to the different

ways of carriers. Lentiviral particle-delivering system is expected to have influence on tumor immunity compared to PX458 (Cas9 may be also problematic). There are just too many variables to make any solid conclusions here.

We show that the two tumour cell-lines grow at identical rates in C57BL/6 mice in the time-frame that we use them (Figure 2G). This experiment is internally controlled in that each mouse is injected with both lines simultaneously, and we see similar growth of the two tumors in each mouse. We also show that they grow at a similar rate *in vitro* (Extended data figure 3I). In this model, it therefore does not seem that their growth is affected by which genetic editing method we use. The KO cell-line does not express Cas9, as is explained in the method section and text: cells were transiently transfected with Cas9 and guide-RNA containing plasmids, which is then diluted out over subsequent cell-divisions before being used *in vivo*. As for the lentiviral transduction of these cells, the only part that is integrated into the genomes, is the DNA between the 5' LTR and the 3' LTS of the transfer vector. This DNA contains the murine sequence for SEMA3A and mCherry, and to our knowledge mCherry has not been shown to be immunogenic in C57BL/6 mice. In any chance, if this was an issue, one would expect to see faster rejection of tumours, not slower, the opposite of what we see here. The main variable that is changed between these two cell-lines is therefore their expression of SEMA3A.

Minor concerns:

1) Figures 6D, 6F, the last column has no label/legend

Fixed.

2) Line 281: "non-cognate H-2 Kd-gp33 H-2 Kb-Ova pMHC". Should H-2 Kb-Ova pMHC be deleted?

Yes, Fixed.

REVIEWERS' COMMENTS

Reviewer #1 (Remarks to the Author):

The authors have answered my comments thoughtfully and comprehensively. Thank you for this. I have no further concerns.

Reviewer #2 (Remarks to the Author):

The authors have adequately responded to my points and added the necessary experiments and discussion. I appreciate the authors' efforts and agree to publish this article.

Reviewer #3 (Remarks to the Author):

Thanks for the thorough revision and thoughtful discussions included. There are some very minor changes required:

- 1) Clearly states mouse genotypes in figure legend (Fig. 2) and text. The current text omitted all CD4-cre and can be misleading.
- 2) Fig. 2F, legend for Nrp1fl/fl (fl/fl is missing)
- 3) Fig. S1E, an arrow for the PlexinA1 and A2 bands will be appreciated.

Concerning Semaphorin 3A causes immune suppression by inducing cytoskeletal paralysis in tumour-specific CD8⁺ T cells

We wish to thank the reviewers for their comments. The effort you have spent on improving this manuscript is deeply appreciated.

Below is attached reviewers' comments in red and our comments in blue.

Comments for reviewers

Reviewer #1: The authors have answered my comments thoughtfully and comprehensively. Thank you for this. I have no further concerns.

Thank you.

Reviewer #2: The authors have adequately responded to my points and added the necessary experiments and discussion. I appreciate the authors' efforts and agree to publish this article.

Thank you.

Reviewer #3: Thanks for the thorough revision and thoughtful discussions included. There are some very minor changes required:

Thank you.

1) Clearly states mouse genotypes in figure legend (Fig. 2) and text. The current text omitted all CD4-cre and can be misleading.

This has been corrected throughout the entire manuscript and in figure legends.

2) Fig. 2F, legend for Nrp1^{fl/fl} (fl/fl is missing)

Flow/flox was missing from the figure of the experimental design. This has been corrected.

3) Fig. S1E, an arrow for the PlexinA1 and A2 bands will be appreciated.

These have now been added.